# ~~RwR:~~ A Reason-while-Retrieve framework for Reasoning on Scene Graphs with LLMs

## Abstract

Large Language Models (LLMs) have demonstrated impressive reasoning and planning capacities, yet grounding these abilities to a specific environment remains challenging. Recently, there has been a growing interest in representing environments as scene graphs for LLMs, due to their serializable format, scalability to large environments, and flexibility in incorporating diverse semantic and spatial information for various downstream tasks. Despite the success of prompting graphs as text, existing methods suffer from hallucinations with large graph inputs and limitation in solving complex spatial problems, restricting their application beyond simple object search tasks. In this work, we explore grounding LLM reasoning in the environment through the *scene graph schema*. We propose *SG-RwR*, an iterative reason-while-retrieve scene graph reasoning framework involving two cooperative schema-guided code-writing LLMs: a (1) *Reasoner* for task planning and information querying, and a (2) *Retriever* for extracting graph information based on these queries. This cooperation facilitates focused attention on task-relevant graph information and enables sequential reasoning on the graph essential for complex tasks. Additionally, the code-writing design allows for the use of tools to solve problems beyond the capacity of LLMs, which further enhance its reasoning ability on scene graphs. We also demonstrate that our framework can benefit from task-level few-shot examples, even in the absence of agent-level demonstrations, thereby enabling in-context learning without data collection overhead. Through experiments in multiple simulation environments, we show that *SG-RwR* surpasses existing LLM-based approaches in numerical Q&A and planning tasks.

## 1 Introduction

Large language Models (LLMs) have shown remarkable prowess in not only language interpretation (Achiam et al., 2023; Touvron et al., 2023) but also reasoning and planning (Song et al., 2023; Zeng et al., 2022). Prior works have successfully leveraged the world knowledge encapsulated in LLMs for plan generation (Song et al., 2023), interaction (Joublin et al., 2024), and action selection (Rana et al., 2023), which suggests a promising path towards embodied intelligence (Huang et al., 2023a; 2022).

Despite much progress, the challenge of grounding the reasoning process of LLMs to situated environments remains unsolved, predominantly due to the absence of a generalizable and explicit representation of environmental spatial and semantic information that LLMs can process (Huang et al., 2023c). One vein of research explores leveraging LLMs to interface with external tools for the extraction of task-oriented states from perceptual data (Liang et al., 2023; Huang et al., 2023b). Although this strategy has shown effectiveness for several manipulation and planning tasks, it requires LLMs to compose tools in a predetermined way taught through in-context learning (Brown et al., 2020), restricting LLMs from reasoning flexibly on novel tasks. Furthermore, sensory inputs such as images capture only a fraction of the environmental information and are inadequate for tasks necessitating a comprehensive understanding of a 3D scene. In contrast, scene graphs have emerged as a powerful and scalable high-level representation of environments Hughes et al. (2022); Gu et al. (2024). Unlike images, scene graphs explicitly encapsulate spatial relationships and offer the flexibility to incorporate diverse semantic and quantitative attributes (Zhu et al., 2021). Additionally, they are parsable by LLMs, thus enabling the direct grounding of LLM reasoning to the underlying environment (Rana et al., 2023; Ni et al., 2023).

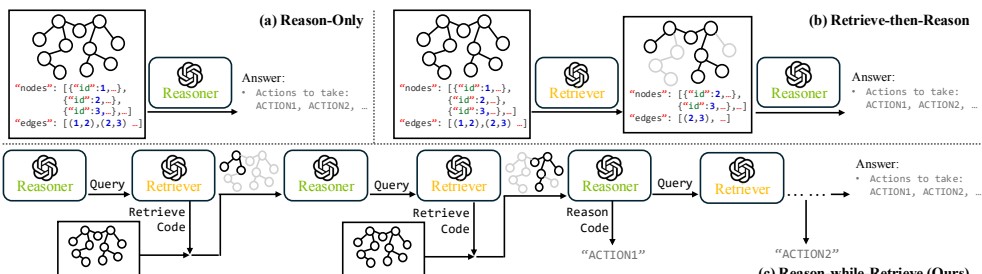

Figure 1: **LLM Graph Processing Framework Comparison**. (a) Reason-Only: A Reasoner LLM is directly prompted with a full textualized graph. (b) Retrieve-then-Reason: A Retriever LLM filters out a task-related sub-graph for use by another Reasoner LLM as text inputs. (c) Reason-while-Retrieve (Ours): A Reasoner and a Retriever collaborate in solving a task by attending to the graph dynamically based on the progress in solving the task. Both Retriever and Reasoner LLMs write code to process information to avoid hallucinations and to enhance numerical and spatial reasoning.

Leveraging LLMs for reasoning with scene graphs remains an under-explored problem. Reasoning requires LLMs to interpret task descriptions, comprehend the relational and semantic information within the graph, and apply their intrinsic knowledge to solve the task by grounding on the graph and in turn the environment. Recent research explores graphs-as-text as the input for LLMs Fatemi et al. (2024); Gu et al. (2024). LLMs are shown to possess a preliminary capacity to interpret graph topology. Yet, they are prone to hallucinations or exceed input token limits when handling large graphs (Wang et al., 2023). To tackle the challenges, (Luo et al., 2024) propose a "Retrieve-then-Reason" wherein the LLM first explores the graph identifying the sub-graph pertinent to a given task, and then performs reasoning on the retrieved part to generate the task solution. The exploration phase employs a heuristic strategy, either by exploring neighborhood nodes and edges of visited parts Sun et al. (2023) or expanding the sub-tree rooted at nodes at a certain hierarchical level Rana et al. (2023). This strategy is adept at information collection, however, it is less suited for intricate tasks that require a comprehensive understanding of the entire graph. It is also limited in its ability to dynamically shift focus based on the reasoning process and the requirements of task sub-steps. Additionally, LLMs are incapable of solving complex spatial reasoning tasks that human experts can solve with ease, due to their well-established limitation in numerical reasoning ability (Nezhurina et al., 2024; Ahn et al., 2024). The aforementioned limitations restrict the utility of LLMs in understanding complex scenes from textualized graphs.

Recent research on interleaved generation and retrieve methods (Yao et al., 2022; Jiang et al., 2023; Press et al., 2022) highlights their advantages over single-time retrieval strategies. By retrieving multiple times, these methods reduce factual errors in LLM responses by iteratively aggregating relevant information throughout the reasoning process. However, adapting them for scene-graph-based information source is not straightforward. Originally designed for reasoning on text corpora, these methods leverage search engines to retrieve sentences or paragraphs that are semantically "close" to the past reasoning context using lexical (Trivedi et al., 2022) or neural embedding analysis (Shao et al., 2023). In contrast, the information required for spatial reasoning tasks considered in this paper demands both semantic and structural understanding of the scene graphs. Failure to capture the spatial relationships can widen the gap between the retrieved information and what is needed for reasoning, ultimately reducing task performance.

In this work, we propose *SG-RwR*, a **S**cene-**G**raph-tailored **R**eason-**w**hile-**R**etrieve framework, depicted in Figure 1. This framework interleaves the reasoning and scene graph information retrieval phases, which ensures that LLMs focus only on the information that is selectively aligned with the task solving process, and that the reasoning trace is grounded in the graph by factoring in the retrieved graph information. Our framework consists of two cooperative LLM-powered modules: a *Reasoner* that decomposes the task and generates queries for the information that can guide subsequent steps; and a *Retriever* that processes the queries and *writes code* to retrieve related graph information for the Reasoner. To prevent hallucinations when processing excessive information, we prompt both LLMs with only the *graph schema* instead of the entire graph. The schema describes the types, format, and semantics of the scene information in the grap. It guides the Reasoner to determine *what* information is helpful to solve a given task, and informs the Retriever to *write code* for accessing the graph as a

database to obtain the desired information. We also equip the Reasoner with code-writing capabilities to conduct precise numerical reasoning (Lyu et al., 2023) and employ external tools for well-defined atomic problems, thereby enhancing the framework's ability to tackle complex scene understanding and planning tasks.

We evaluate our method with two simulation environments: BabyAI (Chevalier-Boisvert et al., 2018), a 2D grid world environment; and VirtualHome (Puig et al., 2018), a large-scale indoor multi-room environment. Our experiments on numerical Q&A and planning tasks show that *SG-RwR* greatly improves the reasoning ability of LLMs on scene graphs. We also observe that *SG-RwR* can effectively leverage end-to-end task-level few-shot examples *without requiring module-level demonstrations*. Additionally, compared to direct graph prompting methods, *SG-RwR* can better extrapolate from few-shot examples to unseen tasks without suffering from severe performance degradation. Specifically, on the traversal plan generation task in BabyAI, our method outperforms baselines by 18.5 percentage points (pp) in the zero-shot prompt setting, and by 3pp and 60pp in seen and unseen environments in the few-shot prompt setting.

In summary, our contributions include:

- An iterative Reason-while-Retrieve (*SG-RwR*) framework with reasoning-oriented information gathering mechanism for task solving on scene graphs.
- Schema-based grounding and code-writing for graph information retrieval and processing that reduces hallucination and improves the reasoning ability of LLMs on complex tasks.
- We show that *SG-RwR* significantly enhances the performance in two distinct environments, encompassing a wide range of tasks in both zero-shot and few-shot settings.

## 2 METHOD

### 2.1 PROBLEM STATEMENT

Our problem setting involves a natural language task instruction $I$ and a scene graph $\mathcal{G} = (V, E)$, where $V$ and $E$ denote vertices and edges, respectively. Each node $V_i$ represents an object along with its attributes, such as coordinates or colors, while each edge indicates a type of spatial relationship, such as inside or on top of. Additionally, we assume access to the *scene graph schema* $\mathcal{S}$, which is a textual description of vertex, edge, and attribute types, formats, and semantics. Our objective is to generate the solution of $I$ using LLMs, based on the available information above, expressed as $\mathcal{A} = f(I, \mathcal{G}, \mathcal{S}; LLMs)$.

### 2.2 OVERVIEW OF *SG-RwR*

While existing methods directly prompt LLMs with textualized graphs, we explore grounding the reasoning process to scene graphs based on the scene graph schema $\mathcal{S}$ and the code-writing ability of LLMs. We develop *SG-RwR*, an LLM-based multi-agent framework that iteratively reasons through the next steps and retrieves necessary information from the graph. As shown in Figure 2, our method contains two LLM agents: a *Reasoner* and a *Retriever*. Given a task, the Reasoner determines the next substep to approach the task and identifies the scene graph information necessary for it. It then raises a natural language query to the Retriever for this information. Upon receiving the query, the Retriever processes the scene graph through code-writing and sends the data back to Reasoner. By iteratively performing these steps, both agents collaborate to solve the task.

Our system initializes with the Scene Graph Schema, the Environment Description, general Guidance to direct the cooperation process, and task-dependent information such as the description of Agent Actions and Reasoning Tools. Then, given the Task, the Reasoner outputs analysis in natural language labeled as *Explanation*, and *Query* the Retriever. In turn, given the Scene Graph and a Query, the Retriever provides structured responses grounded in the Scene Graph. This process iterates until the Reasoner outputs a plan.

The next two subsections explain workflows of each agent, as well as techniques that ensure a fluent and automated task-solving process.

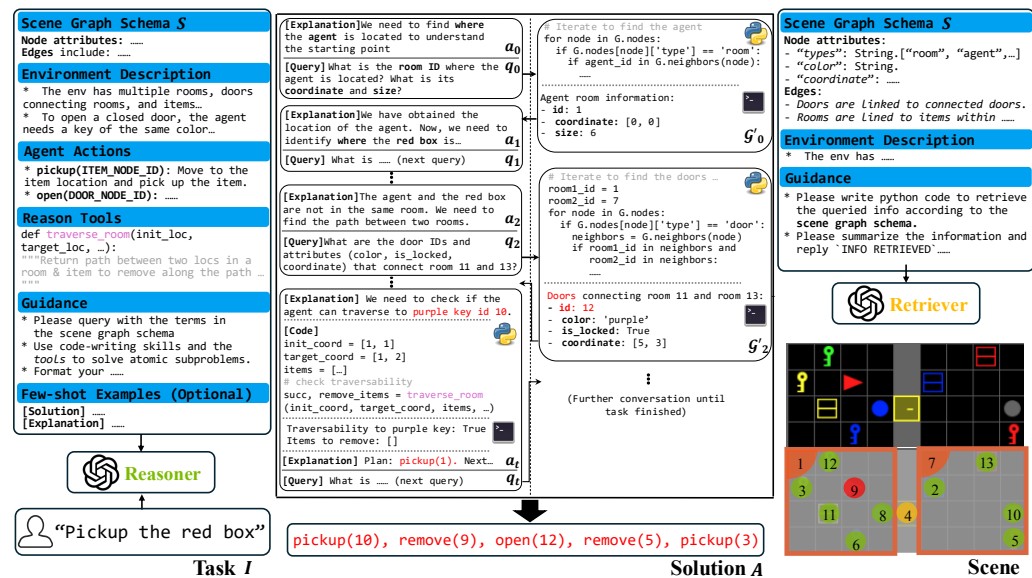

Figure 2: **SG-RwR** **Workflow**. It solves tasks based on scene graphs through the cooperation of two LLM agents: Reasoner and Retriever. Reasoner iteratively queries Retriever for graph information and reasons based on the received data from the Retriever. Additionally, both agents employ the code-writing skill: Retriever writes code to retrieve graph information, while the Reasoner writes code to utilize external tools for solving complex atomic problems. In the graph, and represent code writing and execution stage, respectively. They retrieve graph information $\mathcal{G}'$ or enhance the analysis stages $a$.

## 2.3 REASONER

Reasoner is the core of *SG-RwR*, steering the task-solving iterations. We prompt it with the schema $\mathcal{S}$, environment and task information (such as action description for the planning task), annotations of reasoning tools, general guidance to ensure automated task-solving conversation, and optionally, few-shot task-level examples. Reasoner then initiates the conversation with Retriever to solve a given task.

Concretely, without any knowledge about the graph data initially, the Reasoner analyzes the task $I$ and graph schema $\mathcal{S}$, and generates the first analysis, denoted as $a_0$, and sends out the first associated information retrieval query, designated as $q_0$, to the Retriever to access the graph information. At the $t^{th}$ round of conversation, the Reasoner consumes the conversation history, which includes past information retrieval queries, retrieved information, and the past analyses: $\{(a_0, q_0, \mathcal{G}'_0), \cdots, (a_{t-1}, q_{t-1}, \mathcal{G}'_{t-1})\}$. It then generates the next corresponding analysis $a_t$ and query $q_t$, where $a_t$ involves intermediate conclusions and the next subtask to be solved, which informs and justifies $q_t$. For example, in the $2^{nd}$ round of conversation shown in Figure 2, Reasoner processes previously retrieved agent and red box room and location $(\{(a_0, q_0, \mathcal{G}'_0), (a_1, q_1, \mathcal{G}'_1)\})$, identifies that the next subtask is to find "the path between two rooms" ($a_2$), and then query for the "door IDs and attributes" that connect two rooms ($q_2$) for solving the subtask. In this way, each reasoning step is grounded to the environment by factoring in the retrieved information, and the graph data processed by LLMs is filtered by the reasoning.

The grounded iterative reasoning above involves solving spatial graph problems, such as navigation and object search. Prior work shows that LLMs give unreliable solutions to quantitative problems (Ahn et al., 2024). To circumvent the deficiency, we follow prior work (Schick et al., 2024; Paranjape et al., 2023) to enable code-writing and tool-use for the Reasoner. We provide programmatic functions to address atomic problems critical to the given task family. As shown in Figure 2, at the $t^{th}$ round of conversation, the Reasoner uses the provided pathfinding tool `traverse_room` to identify obstacles that need to be removed to traverse to the key, a problem beyond the capacity of LLMs. We include tool annotations in the prompt to guide the Reasoner in querying for the information necessary. The

introduction of tools prevents hallucination on complex problems and reduces the burden of LLMs by leveraging known algorithms.

Since the Reasoner controls the iterative process to address a task, it is critical to control its behavior to ensure a smooth flow of the conversation. We control the message exchange between the Reasoner and the Retriever through both prompt guidance and manual interference. Specifically, we prompt the Reasoner with the graph schema and the guidance to `"Communicate using the terms in the graph schema"` to avoid confusion. We also filter out only the next query $q_{t+1}$ to send to the Retriever, removing the analysis $a_t$ and the past conversation. We find that without doing so, the Retriever might attempt to realize all plan steps in the language analysis in the conversation, while omitting the actual desired information, which leads to a failure eventually.

## 2.4 RETRIEVER

The Retriever assists the Reasoner by processing its queries and returning the requested information from the graph. Specifically, given a free-form language query $q$, the Retriever generates code that executes on the scene graph to retrieve the relevant subgraph containing the required information $\mathcal{G}' = (V', E') = h(\mathcal{G})$. Here, $V'$ and $E'$ denote subsets of graph nodes and edges, respectively. While the Reasoner may query for either the entire node or edge or just a subset of their attributes, we use $V'$ and $E'$ as the general representation for either case. Similar to the prompt for the Reasoner, the prompt forthe Retriever includes the environment description, the scene graph schema $\mathcal{S}$, and general guidance. The key difference is that $\mathcal{S}$ guides the Retriever in writing the information retrieval code. Confusion is avoided by ensuring that both agents communicate using the same terms from the schema.

## 2.5 SELF-DEBUGGING AND ERROR PREVENTION IN CODE-WRITING

Even with adequate context, LLMs are not guaranteed to write executable code in a single attempt. Therefore, we introduce a self-debugging mechanism to both the Retriever and the Reasoner to ensure the successful execution of their code (Chen et al., 2024). Specifically, we establish an inner iteration between the code-writing LLM and the code executor. At each round, we prompt the history of attempts, including the initial query $q$, previous programs $h_0, \cdots, h_{i-1}$, and execution outcomes $h_0(\mathcal{G}), \cdots, h_{i-1}(\mathcal{G})$, back to the LLM for review. If execution errors exist, the code-writing LLM corrects the code and repeats the process. Conversely, if the code execution is successful, then the debugging iteration terminates.

What's more, we observe hallucination in the code written by LLMs as prior work (Liu et al., 2024). In our case, the Reasoner might hallucinate about scene information without querying for it from the Retriever. To prevent this, we design a reprompting technique based on keyword detection. Specifically, we detect the keywords *"assuming"* and *"assume"* in the code written by LLMs, and prompt the code back to the Reasoner with the query to remove any assumptions in the code. We observe that the simple technique prevents scene information hallucination in most cases.

## 3 EXPERIMENTAL SETTINGS

We evaluate our methods on a series of numerical Q&A (NumQ&A) and planning tasks within the BabyAI (Chevalier-Boisvert et al., 2018; Chevalier-Boisvert et al., 2023) and VirtualHome (VH) (Puig et al., 2018) environments. Detailed descriptions of these environments are provided in the following subsections. For each environment, we provide a single scene graph schema and environment description that is consistent across all tasks for that environment. Our method then generates solutions grounded in different scene graph instance inputs for each experiment.

Each task in our experiments requires reasoning on both the spatial structure and the semantic information encoded in the graph. We use the success rate as our evaluation metric, where success is defined as either providing the correct answer or achieving the desired outcome in the simulation. In this paper, we use GPT-4o for all methods, including *SG-RwR* and the baselines below. *SG-RwR* process is implemented using AutoGen (Wu et al., 2023).

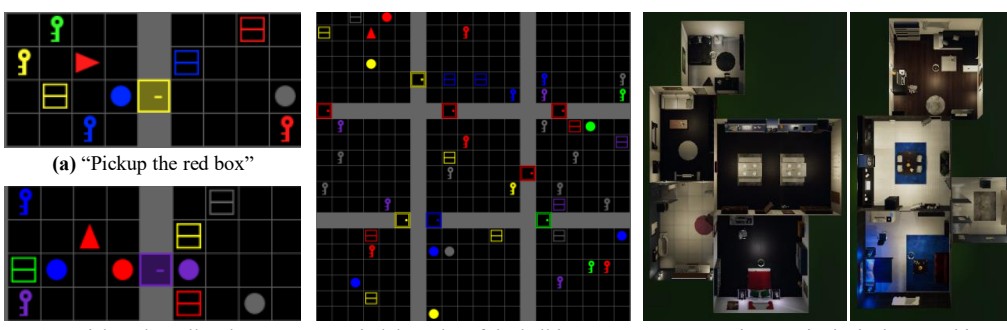

(a) "Pickup the red box"

(b) "Pickup the yellow box"

(c) "Find the color of the ball in a room next to the room with 4 grey keys"

(d) "Put the soap in the bathroomcabinet"

Figure 3: **Experiment Settings**. (Best viewed in color) The environment and tasks for evaluation. **(a)** BabyAI Trv-1 task with single-side door obstacle; **(b)** BabyAI Trv-2 task with double-side door obstacles; **(c)** BabyAI Numerical Q&A task; **(d)** Two VirtualHome household environments (left: VH-1; right: VH-2) and an examplar task.

**Baselines** Following NLGraph (Wang et al., 2023), we compare our approach against several direct reasoning methods based on whole graph prompting. These methods include three zero-shot approaches: **zero-shot prompting** (ZERO-SHOT), **Zero-Shot Chain-of-Thought** (0-COT) (Kojima et al., 2022), **Least-to-Most** (LTM) (Zhou et al., 2022); and three few-shot methods: **Chain-of-Thought** (COT) (Wei et al., 2022), **Build-a-Graph** (BAG) (Wang et al., 2023), **Algorithmic Prompting** (ALGORITHM) (Wang et al., 2023). In addition to the few-shot examples, ALGORITHM also require a language description of the task solving method. We also compare against **ReAct** (Yao et al., 2022), a generic iterative reasoning and acting approach that is able to call database APIs to retrieve information. Furthermore, we compare against **SayPlan** (Rana et al., 2023), a retrieve-then-reason baseline. Compared to other methods of this category, Sayplan is specifically designed on the scene graphs that represent spatial layout, and thus is more suitable for the problem scope considered in this paper. For the detailed function design for SayPlan and ReAct, please refer to Appendix G.

**Few-shot *SG-RwR*** We investigate the performance of *SG-RwR* in both zero-shot and few-shot settings. For the latter, we introduce two few-shot versions of *SG-RwR*: *SG-RwR* **+FewShot**(*SG-RwR*-FS), which incorporates additional in-context learning examples for the Reasoner, and *SG-RwR* **+Algorithm**(*SG-RwR*-A): which adds both in-context examples and algorithmic prompts to the Reasoner. Notably, although *SG-RwR* involves dialogue between two agents, we do not provide either agent with detailed conversation examples, as these can be impractical to collect and may constrain the reasoning flexibility of LLMs. In this way, we examine whether our framework can leverage task-level examples to enhance its reasoning capacity.

## 3.1 2D GRID WORLD NUMERICAL Q&A

Our first experiment is on a numerical Q&A task in a customized 9-room 2D BabyAI (Chevalier-Boisvert et al., 2018) environment, as shown in Figure 3(c). We generate scene graph representation of the environment following the hierachical graph design from 3DSG (Armeni et al., 2019), illustrated in Figure 4. Specifically, the graph represents the spatial scene layout through three levels: root, rooms, and objects, with additional door nodes connecting room pairs.

Inspired by the complex search questions designed in SayPlan (Rana et al., 2023), we design the following question template: `find the color of the {TARGET_OBJECT} in a room next to the room with {NUM_IDENTIFIER} {COLOR_IDENTIFIER} {IDENTIFIER_OBJECT}`, where contents in curley brackets are populated based on each new environment instance. The environment and question pairs are designed to ensure that there is only one answer.

We test each method in 100 different environment and task instance. For few-shot methods, we sample two instances and manually annotate the solution and the explanation as the in-context learning prompt.

## 3.2 2D GRID WORLD TRAVERSAL PLANNING

We also test on the traversal planning task in BabyAI, where the task is to generate a sequence of node-centric actions to pick up a target item. We design three atomic actions, including (1) `pickup(nodeID)`: Walk to and pickup an object specified by the node ID; (2) `remove(nodeID)`: Walk to and remove an object specified by the node ID; (3) `open(nodeID)`: Walk to and open a door specified by the node ID. We directly query *SG-RwR* and all baselines to generate the actions in the format above.

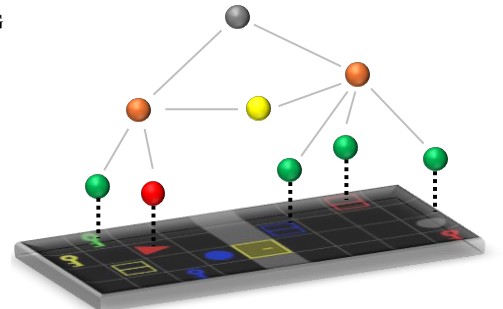

Figure 4: **BabyAI Scene Graph Representation**. Graph nodes represent items, agents, rooms, and doors. Edges indicate items or agents located inside a room, or doors that connect rooms. Room nodes are connected to a root node.

As shown in Figure 3(a)(b), the traversal planning task is tested in two related double-room environments, both of which require the agent to pick up the key of the correct color to unlock the door, remove any obstacle that blocks the door, open the door, and pick up the target. The difference is that the first environment, **Trv1**, contains only the agent-side obstacle, whereas the second environment, dubbed **Trv2**, contains another target-side obstacle. We generate the in-context examples *only* in Trv1, and test if the methods can extrapolate to Trv2. As before, we evaluate each method in 100 times in different instance of both types of the environment. For *SG-RwR*, we provide the reasoning function `traversal_room` programmed based on the $A^*$ algorithm, which identifies the item to remove in order to reach from an initial to a desired location within the same room. As we will show, *SG-RwR* is able to leverage this external tool to compensate for the limited mathematical problem solving ability of LLMs.

## 3.3 HOUSEHOLD TASK PLANNING

Our last evaluation is in two VirtualHome (VH) (Puig et al., 2018) environments shown in Figure 3(d). We denote them as **VH-1** and **VH-2**, respectively. Each of these is encoded as a built-in environment graph that naturally serves as the input to our method. Compared to BabyAI, VH environments are larger in scale in terms of the state space and action space. Both of them contain 115 object instances, 8 relationship types encoded as edges in the graph, and multiple object properties and states that determine the executability of an action. Hence the VH environment is more challenging in terms of task-dependent information distillation. For each environment, we adopt the 10 household tasks from ProgPrompt (Singh et al., 2023), such as `"put the soap in the bathroom cabinet"`, and query each method for the action sequence to accomplish the task. As before, we task each method to directly generate the plan in the VH action format. It includes `[action_name]<object_name>(object_id)` for one argument actions, and `[action_name]<object_name1>(object_id1)<object_name2>(object_id2)` for two argument actions. Two of the tasks, together with the ground truth action sequences, serve as the few-shot examples, whereas the other eight are for testing. To situate the task in the environment, we follow CoELA Zhang et al. (2024) to specify the task as the desired states. For example, the task of above is specified as `soap INSIDE bathroomcabinet`. To achieve the desired state, LLMs need to reason over the current state of the environment in order to identify the sequence of actions that ultimately achieve the achieve the desired outcome. A plan is considered successful if the desired states are reached after simulation. Please refer to Appendix C for more details.

## 4 RESULTS AND ANALYSIS

### 4.1 EXPERIMENT RESULTS

**Numerical Q&A Resutls** The results are collected in Table 1. The vanilla version of our method outperforms the best baseline by 30 percentage points (pp), even though it does not take the advantage of the few-shot examples. In this task, few-shot methods do not show significant advantage over

"Put the soap in the bathroomcabinet"

[walk]<barsoap>(34)  [grab]<barsoap>(34)  [walk]<bathroomcabinet> [open]<bathroomcabinet>  [putin]<barsoap>(34)
                                                    (26)              (26)          <bathroomcabinet>(26)

(a) Retrieve the node with the class_name "barsoap".
(b) We have identified the bar soap with the details:
    - ID: **34**
    - Class Name: barsoap
    - Properties: ['**GRABBABLE**', 'MOVABLE', 'CREAM']

**Reasoner**

(c) Retrieve the node with the class_name "bathroomcabinet".
(d) We have identified the bathroom cabinet with the details:
    - ID: **26**
    - Class Name: bathroomcabinet
    - Properties: ['SURFACES', 'CAN_OPEN', 'CONTAINERS']
    - States: ['**CLOSED**']
(e) We need to **open** the bathroom cabinet (ID: 26) since it is closed.

Figure 5: **VirtualHome Qualitative Demonstration**. **Top row**: Plan Execution; **Middle row**: Generated plan in the VirtualHome action format. **Bottom row**: *SG-RwR* Reasoner-side conversation behind the generated plan.

| | Zero-Shot | | | | Few-Shot | | | | | | |
| | ZeroShot | 0-CoT | LTM | *SG-RwR* | CoT | BAG | Alg | ReAct | SayPlan | *SG-RwR* (FS) | *SG-RwR* (Alg) |
| **Task** | | | | | | | | | | | |
|---|---|---|---|---|---|---|---|---|---|---|---|
| NumQ&A | 55% | 48% | 52% | **95%** | 45% 53% | 51% | 65% | 24% | 35% | **94%** | **97%** |
| Trv-1 | 20% | 23% | 17% | **61%** | 34% | 35% | **64%** | 13 | 18% | **67%** | **64%** |
| Trv-2 | 11% | 7% | 6% | **56%** | 1% | 1% | 0% | 0% | 0% | **61%** | **56%** |

Table 1: **Results in BabyAI** *SG-RwR* achieves the best performance across all tasks in both zero-shot and few-shot settings, showing that *SG-RwR* (1) is effective in solving spatial tasks; (2) can harness the information from in-context examples and extrapolate better to unseen tasks.

zero-shot methods. They can all reason correctly on this problem, but tend to make mistakes when addressing the substeps such as counting the item or locating the neighboring rooms. The room-by-room graph traverse mechanism used in SayPlan further degrades the performance, as the relevance of the information to the task cannot be determined without reasoning first. That is, the target neighboring room cannot be identified without finding the identifier room first. In contrast, *SG-RwR* attends to the graph information in the correct order by querying for it based on the reasoning process.

**2D Traversal Results** Table 1 also reports the success rate for all methods in two traversal environments. In the seen environment, our method achieves 38pp and 3pp higher success rate against the best performing baselines under zero-shot and few-shot settings, respectively. While few-shot baselines perform more than 10pp better compared to zero-shot baselines, they perform even worse in the unseen settings, achieving less than or equal to $1\%$ success rate. This indicates that although few-shot examples help improve the performance in the seen tasks, LLMs do not *learn* the reasoning process to extrapolate to similar unseen tasks. Rather, LLMs might only *memorize* the heuristic mechanism

| Method | Few-Shot Examples | VH-1 | VH-2 |
|---|---|---|---|
| **ZeroShot** | | 87.5% | 75% |
| **0-CoT** | | 87.5% | 75% |
| **LTM** | | 87.5% | 62.5% |
| **CoT** | ✓ | 87.5% | 75% |
| **BAG** | ✓ | 87.5% | 62.5% |
| **RwR** | | **100%** | **100%** |

Table 2: **Results in VirtualHome**. The superior performance of *SG-RwR* shows that it is capable of grounding its plan to the environmental states.

that can help solve the same task, such as removing the item on the left of the door in this case. On the other hand, by separating out the Retriever that handles the graph information, the Reasoner in *SG-RwR* learns the reasoning process from the few-shot examples that is essential for the task, and can thus extrapolate well to similar problems utilizing the knowledge. SayPlan achieves even inferior results compared to reason-only methods, indicating that its heuristic retrieval method is unsuitable for tasks concerning global information.

| Method | Code-Writing & Tool-Use | Iterative process | Numerical Q&A | Trv-1 | Trv-2 |
|---|:---:|:---:|:---:|:---:|:---:|
| **SingleCoder** | ✓ | | 80% | 33% | 25% |
| **RwR_Text** | | ✓ | 57% | 18% | 8% |
| **RwR** | ✓ | ✓ | **95%** | **61%** | **56%** |

Table 3: **Ablation in BabyAI traversal and numerical Q&A**. The best result is achieved by combining both Reason-while-Retrieve framework and the code-writing, justifying the key designs in our method.

**Household Task Planning Results**   The planning success rate on the 8 tasks in the 2 VH environments are shown in Table 2. We observe that all baselines consistently fail to address the precondition of the planned action. For example, all of them failed to generate `[open] <garbagecan> (ID)` before `[putin] <plum> (ID) <garbagecan> (ID)`, forgetting that the state of the garbage can is `state:{CLOSED}` from the extensive graph input. On the other hand, *SG-RwR* doesn't process the entire graph. Instead, it queries for the specific object information, which helps to better determine the action parameter and examine the action preconditions. For qualitatitve demonstration, please refer to Figure 5 for an examplar task and solution by our method.

## 4.2   ABLATION

**Setup**   To further validate the design of *SG-RwR* framework, we conduct an ablation study for the key component of our method. To this end, we introduce two variants of *SG-RwR*:

- **SingleCoder**: A single LLM that directly writes the entire code to address a given task. It benefits from the accurate numerical reasoning and tool-use capacity from the code-writing, but does not have the opportunity to analyze the intermediate graph information from the iterative retrieving and reasoning (dubbed *Iterative RetRea* in this section). We prompt the SingleCoder LLM with the combination of the information for both the Reasoner and the Retriever in *SG-RwR*, including the environment and action space information, scene graph schema, and tool annotations. The self-debugging mechanism is also introduced.

- *SG-RwR* **_Text**: The other variant disables the code-writing ability of both the Retriever and Reasoner in *SG-RwR*. Instead, both cooperative agents rely purely on language reasoning and communication skill to solve a given task. This design evaluates the performance of the iterative retrieve and reason process without the code-writing. We observe that this variant is only capable of generating plans in natural language. Hence, we add an additional action translator that converts the output to the executable action format, following prior works (Song et al., 2023; Huang et al., 2023c).

Both variants are tested in BabyAI Trv-1 and Numerical Q&A tasks under the zero-shot setting.

**Results**   The ablation study results are demonstrated in Table 3, where both variants impair the effectiveness of the method. While the iterative task solving can better break the task down, correct solution for each substep cannot always be obtained without the code-writing. For example, queried with `"Find all rooms that contain 5 green balls"`, the non-code-writing Retriever is not able to solve the counting problem and locate the correct room without code-writing. On the other hand, while SingleCoder is better at solving numerical problems, it is unable to address complex planning tasks without the iterative cooperation. By combining the advantage from both designs, our method achieves the best result over both variants and all baselines.

## 5   RELATED LITERATURE

**Language models for Task and Motion Planning**   With the advance of large language or multi-modal models, many earlier works look into harnessing their power for decision making (Xi et al., 2023; Chen et al., 2023; Liu et al., 2023) and robotic control (Dalal et al., 2024; Zhang et al., 2023; Lin et al., 2023; Chen et al., 2021; Hatori et al., 2018). With rich built-in knowledge and in-context

learning ability trained from the large internet-scale text corpora, language models are used for generating task-level plans (Raman et al., 2022; Gao et al., 2024), action selection (Ahn et al., 2022; Nasiriany et al., 2024), processing environmental or human feedback (Skreta et al., 2023), training or finetuning language-conditioned policy models (Team et al., 2024; Padalkar et al., 2023; Szot et al., 2023), and more. To allow the language models to factor in the environment during planning, recent studies have explored using LLMs for programmatic plan generation (Singh et al., 2023), combining knowledge from external perception tools via code-writing (Liang et al., 2023; Huang et al., 2023b) or grounded decoding (Huang et al., 2023c), and value function generation Yu et al. (2023). While proven effective, those methods are limited to small scale environments, and rely on multimodal or expert perception models to extract task-related states from the scene representation with implicit spatial structure. In this work, we study using pretrained LLMs to process the the global representation of large environments with explicit structure, and generate the solution that is grounded in the environment.

**Graph as the Scene Representation**  The scope of the solvable task is largely determined by the state representation. Compare to sensory representation such as images or point clouds, scene graphs are compact thus scalable to large environments Greve et al. (2024), structured to represent spatial layout explicitly Hughes et al. (2022); Wu et al. (2021), and efficient in representing diverse states of the environment Armeni et al. (2019). Due to that reason, they have been used in various manipulation or navigation tasks Ravichandran et al. (2022); Zhu et al. (2021). In this paper, we exploit these favorable features of the scene graph representation to ground the reasoning process of LLMs to the environment.

**LLMs for Reasoning on Graph**  Leveraging language models for reasoning on graphs is a growing area. While prior works integrates learnt graph and language knowledge through training or finetuning (Ye et al., 2023; Ni et al., 2023), recent study explore serializing graph-structured data for prompting to pretrained LLMs (Wang et al., 2023; Fatemi et al., 2024). This strategy has been successfully used to enhance the reasoning ability of LLMs with external knowledge graphs (Sun et al., 2023; Luo et al., 2024) and robotic task planning on open vocabulary 3D scene graphs (Gu et al., 2024). Closest to our work, SayPlan (Rana et al., 2023) prompts scene graphs to LLMs and designs a Retrieve-then-Reason framework for the planning tasks. However, it designs the room-by-room retrieve mechanism only for the object search purpose, whereas we design the Reason-while-Retrieve framework that allows graph information retrieval for any type of reasoning. We further incorporate the code-writing and tool-use ability to LLMs, so that our proposed method can effectively retrieve information based on scene graphs and address numerical tasks that fall beyond the expertise of LLMs (Nezhurina et al., 2024).

## 6 CONCLUSION AND FUTURE WORK

In this work, we have proposed *SG-RwR*: an iterative, multi-agent framework that grounds LLMs in a physical environment through scene graphs, and enables them to reason using both natural and, crucially, programming languages. Specifically, *SG-RwR* facilitates reasoning on large scene graphs by enabling LLMs to write code that retrieves task-related information *during* the reasoning process.

Our ablation study shows that both the iterative cooperation process *and* the code-writing design are crucial to the framework's enhanced performance. The former ensures that the data specific to the environment enters the planning process in a just-in-time manner, while later enables prompting with a data *schema* instead directly with the data itself. In short, both of these are ways to limit "information overload" in the Reasoner.

One unexplored benefit of the *SG-RwR* framework is its inherent flexibility: new agents with new specialties can be added to the framework with ease. In future work, we plan to experiment with a third agent, the Verifier, to correct mistakes in the Reasoner's plan based on the graph information. Another promising direction is to add new agent expert on new modalities to integrate richer information about the environment into our method. The iterative nature of *SG-RwR*, however, can lead to longer task-solving times: The number of conversation rounds required increases with task complexity and the number of agents. This suggests future work investigating additional agents must be accompanied with methods to steer the LLMs to minimize the required conversation rounds.

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

## A    PROMPT TEMPLATES FOR *SG-RwR*

*SG-RwR* adopts template-based prompt generation for both the Reasoner and Retriever. The templates for them are shown in Table 4 and Table 5. The prompt is generated by populating the red contents in the template with the specific graph information.

Table 4: Reasoner Prompt Template

---

**Reasoner Prompt Template**

```
You are a planning agent that is excellent at collaboration and
code writing. Given the environment description, schema of the
graph representaiton of that environment, a retriever agent that
can retrieve information from the graph, and a set of user defined
reasoning tool(s), you know what information to ask from the
retriever and how to use them as well as the reasoning tool(s) to
solve a planning task. Then you can generate a plan executable by
the agent to achieve the given mission.

Environment Description:
{ENVIRONMENT PROMPT}

Scene Graph Schema:
{SCENE GRAPH SCHEMA PROMPT}

Agent Actions:
{AGENT ACTIONS}

Please follow the guidance below:
* Solve tasks step-by-step. Figure out the next step that can help
you get closer to the solution.
* If you need any information from the graph based on the graph
schema, raise a language query. A retriever will return the
information to you.
* If you have enough information to solve the next substep, use
your reasoning and code writing skill to solve it. If you write
code, print out the result with succint explanation. The code
execution output will be sent back to you.
* You might be provided with reasoning tools. They are a set of
python functions for solving an atomic subproblem, which might be
helpful for your task. Please use the tools whenever suitable. The
annotation of the tools will be provided at end of the guidance.
* When asking the retriever for information:
    - Raise language queries that are clear, self-contained, and
    addressable by traversing through the graph.
    - Communicate using the terms in the graph schema.
    - Please break questions into simpler queries and raise them
    one-by-one. Avoid asking for all necessary information at once.
* When the task is solved, summarize the solution and reply 'TASK
TERMINATE' in a separate paragraph. Do this ONLY when you obtain
the complete solution.
* Format your information query message in the following way:
    [Explanation]
    Explane why querying for the information.
    [Query]
    The information retrieval query to the retriever.
* Format your code writing message in the following way:
    [Explanation]
    Explain what your code does.
    [Code]
    Python code that solves a subproblem. Wrap the code in the
    python code block.
* Format your entire solution summary message in the following way:
    [Summary]
    Summarize the enire solving process.
    [Actions]
    [ACTION1, ACTION2, ...]
    TASK TERMINATE
```

Table 5: Retriever Prompt Template

---

**Retriever Prompt Template**

```
You are a excellent graph information retrieval agent. Given the
environnment description and the schema of the graph
representaiton of the environment, you are good at writing code to
obtain information from a graph following language queries.
Environment Description:
{ENVIRONMENT PROMPT}

Scene Graph Schema:
{SCENE GRAPH SCHEMA PROMPT}

Please follow the guidance below:
* Please write python code to retrieve information from the graph.
Please include node id in your result and print out the result in
your code.
* If there is no required information stored in the graph, print
None in your code.
* The code execution result will be send back to you. Please check
the result. If the information is retrieved, summarize the
information and replay 'INFO RETRIEVED' in a separate paragraph
following the format below:
    [Summary]
    Summarize the required information
    INFO RETRIEVED
```

---

## B  BABYAI ENVIRONMENT AND SCENE GRAPH DETAILS

**Node attributes**   The node attributes in BabyAI scene graph involve:

- **"type"**: String. The type of the element type. Choices:

$$root, room, agent, key, door, box, ball$$

- **"color"**: String. For doors and items. The color of the element.
- **"coordinate"**: List of integer. Exist for all types of nodes except for the root node. For room nodes, the top left corner coordinate. For other nodes, the 2D coordinate in the grid.
- **"is_locked"**: Binary. For door. State indicating if a door is locked or not.
- **"size"**: List of integer. For room. The size of a room.

## C  VIRTUALHOME ENVIRONMENT AND SCENE GRAPH DETAILS

**Node attributes**   The node attributes in VirtualHome involve:

- **'id'**: Int. Node id.
- **'category'**: Str. Meta category. E.g. "Room".
- **'class_name'**: Str. Specific class name. E.g. "bathroom".
- **'prefab_name'**: Str. Instance name.
- **'obj_transform'**: Dict. 'position': 3D vector, 'rotation': Quaternion form as 4D vector, 'scale': 3D vector
- **'bounding_box'**: Dict. 'center': 3D vector, "size": 3D vector
- **'properties'**: List. Object properties. Determine the action that can act upon it.
- **'states'**: List. Object states. Full list of available states: ['CLOSED', 'OPEN', 'ON', 'OFF', 'SITTING', 'DIRTY', 'CLEAN', 'LYING', 'PLUGGED_IN', 'PLUGGED_OUT', 'HEATED', 'WASHED']

**Edge attributes**    The edge attributes in VirtualHome involve:

- **'from_id'**: Int. Id of node in the from relationship.

- **'to_id'**: Int. Id of node in the to relationship.

- **'relationships'**: Str. Relationship between the 2 objects. Available relationships:

    – **'ON'**: Object from_id is on top of object to_id.
    – **'INSIDE'**: Object from_id is inside of object to_id.
    – **'BETWEEN'**: Used for doors. Door connects with room to_id.
    – **'CLOSE'**: Object from_id is close to object to_id (< 1.5 metres).
    – **'FACING'**: Object to_id is visible from objects from_id and distance is < 5 metres. If object1 is a sofa or a chair it should also be turned towards object2.
    – **'HOLDS_RH'**: Character from_id holds object to_id with the right hand.
    – **'HOLD_LH'**: Character from_id holds object to_id with the left hand.
    – **'SITTING'**: Character from_id is sitting in object to_id.

**Action Space**

- **[walk] <class_name> (id)**: Walk to an object.

- **[grab] <class_name> (id)**: Grab an object. Requires that the agent has walked to that object first.

- **[open] <class_name> (id)**: Open an object. Requires that the agent has walked to that object first.

- **[close] <class_name> (id)**: Close an object. Requires that the agent has walked to that object first.

- **[switchon] <class_name> (id)**: Turn an object on. Requires that the agent has walked to that object first.

- **[switchoff] <class_name> (id)**: Turn an object off. Requires that the agent has walked to that object first.

- **[sit] <class_name> (id)**: Sit on an object. Requires that the agent has walked to that object first.

- **[putin] <class_name1> (id1) <class_name2> (id1)**: Put object 1 inside object 2. Requires that the agent is holding object 1 and has walked to the object 2.

- **[putback] <class_name1> (id1) <class_name2> (id1)**: Put object 1 on object 2. Requires that the agent is holding object 1 and has walked to the object 2.

**Example Task and State-based Specifications in VH-1**    We show the 5 example tasks and their desired final state in the VH-1 environment in Table 6.

| Task Name | State Specification |
| --- | --- |
| Watch TV | tv ON |
| Turn off tablelamp | tablelamp OFF |
| put the soap in the bathroomcabinet | barsoap INSDIE bathroomcabinet |
| throw away plum | plum INSIDE garbagecan |
| make toast | breadslice INSIDE toaster; breadslice HEATED |

Table 6: **Results in BabyAI** *SG-RwR* achieves the best performance across all tasks in both zero-shot and few-shot settings, showing that *SG-RwR* (1) is effective in solving spatial tasks; (2) can harness the information from in-context examples and extrapolate better to unseen tasks.

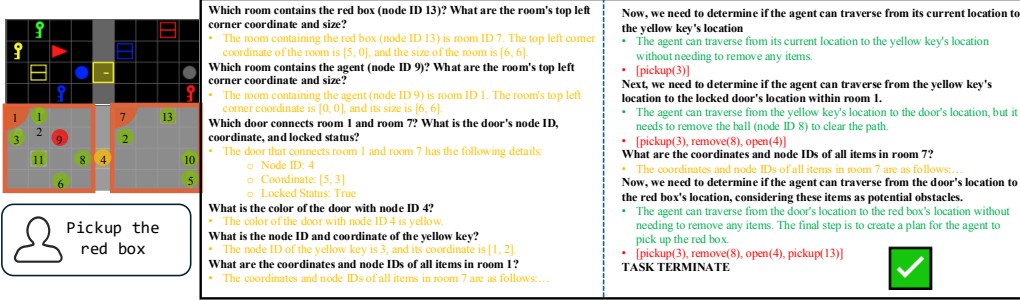

Figure 6: **Example *SG-RwR* Traversal task solving process (Reasoner-side)**. It shows the queries or analysis generated by the Reasoner (in black), information obtained from the Retriever (in yellow), the intermediate conclusion obtained through code-writing that processes the graph information (in green), and the derived plan (in red). The final plan can successfully achieve the mission shown on the left.

## D    EXAMPLE *SG-RwR* COOPERATION ON BABYAI TRAVERSAL TASK

We qualitatively demonstrate how *SG-RwR* addresses a challenging BabyAI traversal task in Figure 6. It shows the task solving process from the Reasoner's perspective, including the information queried from the Retriever as well as the intermediate solution obtained through its own code writing. It clearly demonstrates that *SG-RwR* is able to ground the plan to the environment by iteratively retrieving graph information based on the task solving process and establishing the next step towards solution based on the past retrieved information.

## E    ANALYSIS ON THE COMPUTATIONAL COST

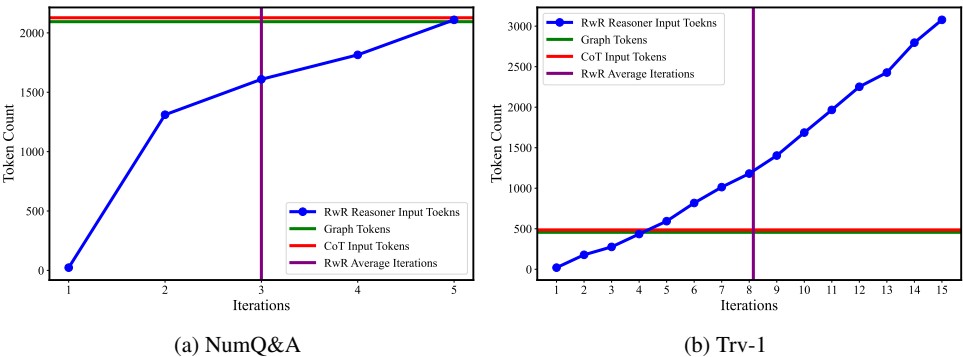

(a) NumQ&A

(b) Trv-1

Figure 7: **Compute Analysis**. We show average conversation rounds and processed token counts at each iteration by *SG-RwR* Reasoner for both NumQ&A (left) and Trv-1 (right) tasks. We also demonstrate the average token counts of the textualized environment scene graph and CoT input for reference.

We show the number of the token processed by our method by iterations and average conversation rounds required to solve a query for the BabyAI tasks in Figure 7. We also plot the token counts of the scene graph and the CoT baseline input. As a direct whole-graph prompting method, the compute required by CoT is determined by the graph size. So the processed token for NumQ&A is 4 times larger than that for the Trv-1, despite that the former is a simpler task requiring less reasoning steps.

On the other hand, *SG-RwR* processed token number monotonically increase along the iteration, as it processes the cumulative conversation history. Hence, the compute required by *SG-RwR* also depends on the task difficulty. However, thanks to the code-writing-based retrieval design, *SG-RwR* only processes limited tokens in early iterations. Thus, for simpler task such as NumQ&A, *SG-RwR*

process less tokens compared to graph prompting method such as CoT at each iteration, which is helpful for reducing hallucination over redundant information. For the traversal task, the processed token count of our method grows beyond even the graph size. This trade-off in compute cost yields superior performance, as demonstrated in Table 1.

## F    How *SG-RwR* avoids hallucination via Code-writing

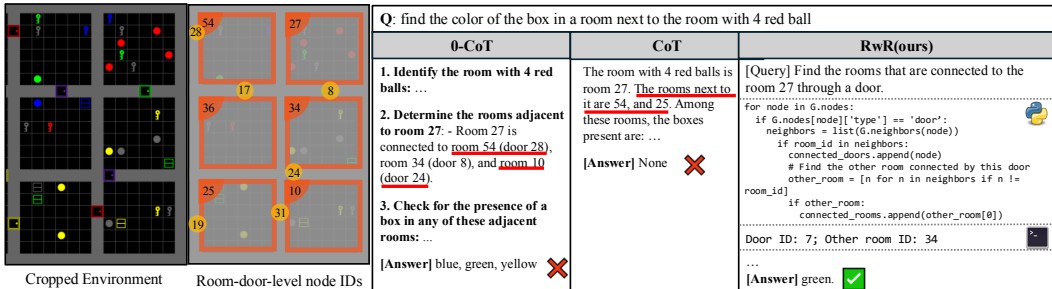

(a) Baseline Hallucination: Spatial layout understanding.

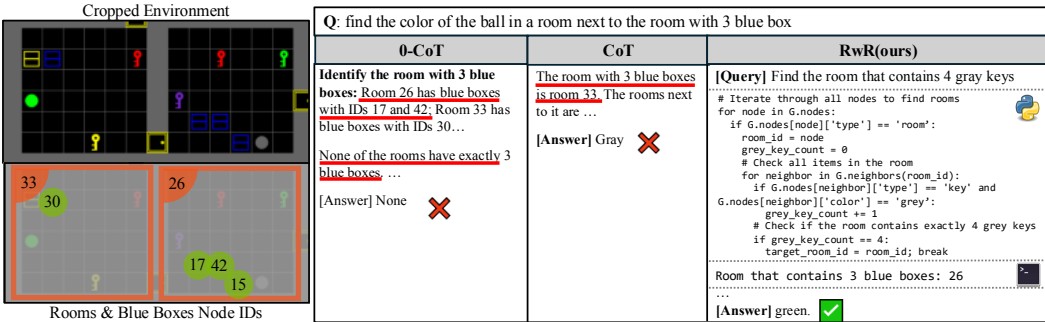

(b) Baseline Hallucination: Spatial counting problem.

Figure 8: **Qualitative demonstration on how *SG-RwR* avoids hallucination.** We show how baselines might hallucinate under the following subtasks: **(a)** Interpreting spatial layout from the scene graphs, where they identify the incorrect neighbor rooms; **(b)** Addressing the counting problem under spatial constraint, where they miscount the number of a target item type in the room. *SG-RwR* is able to avoid the hallucination via code-writing, which filters and processes the graph information more reliably.

In Figure 8, we qualitative show how *SG-RwR* avoids hallucination problems happened on baselines under several scenarios from our tasks. We use the zero-shot **0-CoT** and the few-shot **CoT** as comparison. To focus on the key difference, we only show snippets of reasoning processes for each referent subtask. We show that when reasoning in language, baselines have the tendency to hallucinate in the interpretation of the spatial layout from the scene graph structure, and in address simple quantitative reasoning (e.g. counting) tasks. On the other hand, based on the scene graph schema understanding, *SG-RwR* is able to solve these subtasks more reliably via code-writing.

## G    Baseline Details

### G.1    ReAct

For ReAct, we create the following graph information retrieval APIs in the list below. Each of them is a wrapper of a basic NetworkX (Hagberg et al., 2008) operation:

- **`get_nodes()`**: Get all node IDs in the scene graph.
- **`get_links()`**: Get all links in the scene graph.
- **`get_attrs(node_id)`**: Get the all attributes of a target node;
- **`get_neighbors(node_id)`**: get all neighbor node IDs of a target node.

## G.2 SAYPLAN

SayPlan (Rana et al., 2023) is tested in BabyAI tasks. We follow the original work to create the following APIs for the room-level graph traversal purpose:

- **`collapse(`**$\mathcal{G}$**`)`** for retaining only room and root nodes;
- **`expand(node_id)`** for revealing all nodes rooted from a given room node;
- **`contract(node_id)`** for removing all nodes rooted from a given room node;

We don't assume a graph simulator available for validating and refining the solution as is done in the original paper. Instead, we evaluate the LLM-generated plan by executing it directly in the BabyAI.

