# OpenReview forum: "$\textit{RwR}$: A Reason-while-Retrieve framework for Reasoning on Scene Graphs with LLMs"
_ICLR.cc/2025/Conference — Submitted to ICLR 2025_

### Official Review · Reviewer_chDs · 2024-10-28

**Soundness:** 3
**Presentation:** 2
**Contribution:** 3
**Rating:** 5
**Confidence:** 3

**Summary:**

The paper introduces a cooperative graph reasoning framework called **Reason-while-Retrieve** (**RwR**) , designed to enhance the capabilities of large language models (LLMs) across two simulation environments, BabyAI and VirtualHome. Unlike previous approaches, RwR leverages the collaboration of two specialized agents—a Reasoner and a Retriever —which write code to process textual graph-based information, mitigating the effect of hallucinations. Furthermore, the proposed method achieves state-of-the-art performance on tasks in BabyAI, Numerical Q&A, and VirtualHome.

**Strengths:**

The topic of this paper, leveraging LLMs for reasoning with scene graphs, is both interesting and significant. The strengths of this work are as follows:

1. The figures are clearly presented, making it easy to follow and understand the message the authors are conveying.

2. The paper introduces code-writing and schema-based grounding techniques for graph reasoning, which effectively help mitigate hallucinations in LLM reasoning.

3. The proposed approach, RwR, significantly enhances the performance of LLMs in tasks like Numerical Q&A, 2D Traversal, and Household Task Planning. Notably, it achieves a perfect score (100%) in the VirtualHome environment.

**Weaknesses:**

The weaknesses are illustrated as below:

1. **Lack of Novelty**: The core idea presented in the paper does not appear sufficiently novel. While the paper introduces an iterative reasoning process, the concepts of a "Retriever" and "Reasoner" are already well-established in the field of graph reasoning. Most importantly, the authors fail to provide detailed experimental results regarding how many rounds of iteration the RwR requires, which weakens the contribution of the proposed method.

2. **Incomplete Experimental Results**: Some experiment results seem not complete, such as RwR-FS and RwR-A on the NumQ&A dataset in Table 1.

3. **Lack of Evaluation on Hallucinations**: Given that one of the core motivations of the paper is to mitigate hallucinations in graph reasoning, it is essential to provide some relevant experimental results, like hallucination rates or detailed case studies. This would allow readers to better understand the effectiveness of the proposed method in addressing this issue.

**Questions:**

In addition to the weaknesses previously highlighted, could the authors consider providing results using an open-source LLM? This would offer greater clarity and accessibility for the research community.

---

> ### Author Response · Authors · 2024-11-20
>
> We kindly refer the reviewer to the general response for the following issues:
> 1. What is the novelty of the work compared to ReAct.
> 2. The analysis on the increased inference time due to the iterative conversation process.
> 3. Complete results on the few-shot settings.
> Below we clarify the other concerns.
>
> ## Q1) Questionable novelty since the concepts of “Retriever” and “Reasoner” are already well-established in the graph reasoning field.
> We agree with this and appreciate the reviewer for pointing it out. As explained in the general response, we do not claim novelty in inventing “retriever” or “reasoner” or the interleaving process. Rather, our contribution lies in the designs of those components for the structured scene graph data. We update our title to exclude “RwR” to de-emphasize those concepts. We also change the abbreviation of our method from *RwR* to *SG-RwR* to clarify our contribution. Please refer to the general response for the detailed discussion on the novelty of our work.
>
> ## Q2) Evaluation on the hallucination issues and how SG-RwR mitigates those.
> We add qualitative demonstrations on how SG-RwR mitigates hallucination issues by the baselines in **Appendix F and Figure 8**. These show the scenarios where interpreting scene graphs in language is less reliable than via code, which motivates our design. Such scenarios include: (1) Identifying the neighborhood rooms, which requires understanding the graph links and nodes of a particular type; (2) Obtaining the number of a type of item in a room, which requires not only understanding the graph structure and node semantic attributes, but also address the counting problem.
>
> ## Q3) Can the open-source LLMs be used in the SG-RwR framework?
> We would like to add a placeholder for this issue here. Conceptually our method can be used with any open-source LLMs. However, our implementation is based on the AutoGen library, which is majorly developed OpenAI models.  We are exploring implementation strategies to better integrate with open-source models such as Llama 3, and will update the results here if it can be finished within the discussion period. Otherwise, we could report the performance in the appendix if the paper were to be accepted.

---

> ### Author Response · Authors · 2024-11-25
> **Looking Forward to Your Reply**
>
> Dear Reviewer chDs,
>
> Thank you for your time to read and review our paper. We have followed your suggestions to complete the experiment results and provide qualitative demonstrations on how our approach avoid hallucinations. We have also discussed in length the novelty and the positioning of our work, shown in both the general response and the discussion with Reviewer MeSM and Tptn.
>
> As the discussion period approaches the end, we would appreciate if you could review our response. Please feel free to reach out if you have any additional questions.
>
> Thanks,
> Authors

---

> > ### Author Response · Authors · 2024-12-02
> > **Looking forward to your reply!**
> >
> > Dear Reviewer chDs,
> >
> > We would like to thank you again for the suggestions and insights you provided. Following those, we have clarified the contribution of our method in the context of scene graph reasoning, explained the difference between our methods and more baselines with "retriever" and "reasoner" (ReAct), and showed the advantage of our method via experiments and ablations. We also show how the key designs of our method avoid hallucinations ( Appendix F and general response).
> >
> > As the discussion period approaches the end, we would appreciate if you could review our response. Please feel free to reach out if you have any additional questions.
> >
> > Thanks,
> >
> > Authors

---

### Official Review · Reviewer_Tptn · 2024-10-31

**Soundness:** 3
**Presentation:** 2
**Contribution:** 2
**Rating:** 5
**Confidence:** 4

**Summary:**

The paper introduces Reason-while-Retrieve (RwR), a framework that grounds large language models (LLMs) in environments represented as scene graphs to enhance complex reasoning. RwR uses two cooperating LLM agents: a Reasoner for task planning and a Retriever for extracting relevant subgraphs, working in an interleaved manner to overcome information overload in the Reasoner. This framework also enables tool-assisted problem-solving through code-writing. Experiments demonstrate that RwR outperforms existing methods across several tasks.

**Strengths:**

1. The proposed Reason-while-Retrieve (RwR) framework introduces an interleaved approach, alternating between subtask planning and targeted subgraph extraction from scene graphs, enhancing reasoning performance.

2.  The schema-based grounding and code-writing enables precise information retrieval from graphs, minimizing hallucinations and strengthening LLM reasoning on complex tasks.

3. Experimental results indicate that RwR significantly improves performance across various tasks in distinct environments.

**Weaknesses:**

1. The novelty of the Reason-while-Retrieve (RwR) framework is somewhat limited, as it closely resembles the ReAct framework. ReAct also utilizes an interleaved approach where LLMs generate both reasoning traces and task-specific actions to address hallucination issues. In each iteration, ReAct first produces a reasoning "thought" and then generates an action to interface with and gather information from external environments.

2. The procedural details of RwR are not fully articulated. As shown in Figure 2, some iterations appear to exclude the query step, leaving it unclear how these missed queries are managed within the framework.

**Questions:**

1. What are the key differences between RwR and ReAct? Could the authors consider using ReAct as a baseline for comparison?

2. In Table 1, for the NumQ&A task, why does ZeroShot outperform CoT methods (both 0-CoT and Few-Shot CoT)?

3. Typos:
   - In the middle of Figure 2, there are two instances of *a_t*.
   - In line 192, it’s unclear why the conversation history includes *(q_t, G_t)*, and the definition of *G_t* is missing.
   - In line 197, it states "query for the 'door IDs and attributes' that connect two rooms (a2) for solving the subtask," *a2* should be *q2* here.

---

> ### Author Response · Authors · 2024-11-20
>
> We kindly refer the reviewer to the general response for the following issues:
> 1. What is the novelty of the work compared to ReAct.
> 2. The assumption on the availability of the scene graph schemas and reasoning tools.
> Below we clarify the other concerns.
>
> ## Q1) Ambiguity in Figure 2: some iterations appear to exclude the query step; there are two instances of $a_t$.
> Both ambiguities come from the last two blocks containing code writing and execution result, respectively, from the Reasoner side in Figure 2. They were both labeled as analysis step $a_t$ because, conceptually, this paper treats the explanation before the code writing, code, code execution outcome, and the textual summary of the code outcome (not shown in the original Figure 2) to be in _the same analysis step_. Because they all serve the same purpose:  Solving the subtask raised in the initial explanation, which is identified by the Reasoner to be solvable based on previously collected information. This can be considered as a special analysis step, where the code writing and executor facilitate solving the subtask.
>
> Our original Figure 2 also failed to show that the query is not excluded. Instead, it is generated after the code execution outcome is summarized. **We update the Figure 2 to merge the code writing and execution outcome block into one box, add the missing code outcome analysis and next query raising step, and update the caption to clarify the graph**.
>
> ## Q2) Why does ZeroShot outperform CoT methods on the NumQ&A task in Table 1?
> To make the explanation below more understandable, we would like to first refer the reviewer to the general response about the updated CoT performance. It remains inferior to ZeroShot but with a smaller margin. We would also like to refer the reviewer to the **Appendix F and Figure 8 in the revised manuscript for the qualitative demonstrations on the CoT and 0-CoT failure cases**.
>
> We observe that although the CoT method is able to break the task down correctly, the generated solution for each step is not always reliable. This might lead to the inferior performance of CoT compared to the ZeroShot without task break-down, as an error in any intermediate step might lead to the incorrect final answer. For the NumQ&A task, in order to *“find the color of the box in a room next to the room with 4 red ball”*, CoT method correctly identifies that it need to solve the following two tasks: *“Which room has 4 red balls”*, and *“which rooms are the neighbor of that room”*. However, LLMs are prone to hallucination for such tasks when processing graphs as texts, as shown in the Figure 8.
>
> Additionally, we observe that without few-show examples, 0-CoT tends to break the task down into finer levels. This might even increase the risk of hallucination. For the above example, in order to figure out *“Which room has 4 red balls”*, 0-CoT might attempt to generate the red ball counts in all rooms. In this case, the increased number of subtasks, which are not what LLMs are majorly trained on, might even increase the risk of failure. This might explain the even worse performance of 0-CoT compared to CoT and ZeroShot.
>
> ## Q3) Typos.
> Thanks for pointing those out! We fixed all typos in Figure 2 (see above) and Section 2.3, paragraph 2, so that they correctly reflect the procedural details of our method.

---

> > ### Comment · Reviewer_Tptn · 2024-11-24
> >
> > Thank you for your response. Most of my concerns have been addressed.
> > However, I still have some reservations regarding the ReAct implementation, especially when compared to your work:
> >
> > 1. In terms of retrieval, I agree that the Python implementation is more capable than API calls, and the performance of the ReAct baseline heavily depends on the API design. However, this does not constitute a significant novelty.
> >
> > 2. I do not see a substantial difference between the two-agent framework and the one-agent framework (ReAct). In both frameworks, the reasoner relies on the entire history to generate the analysis and query (reasoning and action in ReAct). The primary distinction is that RwR sends the query to another agent, whereas ReAct executes the action by calling an API.
> >
> > As a result, the performance improvement appears to mainly stem from the use of Python code.

---

> > > ### Author Response · Authors · 2024-11-25
> > >
> > > We appreciate your feedback on our responses, and we would like to provide further details and explanations to justify the novelty of our approach.
> > >
> > > In our two-agent framework, we streamline communication by sending only query sentences to the Retriever, omitting the reasoning behind them, and returning only the summarized retrieved information to the Reasoner, excluding code-writing and debugging history. This isolation of the reason and retrieve process is critical for the success of our method, given the greater autonomy in code writing and execution of Retriever and Reasoner compared to ReAct, which merely suggests APIs in text and relies on manual effort to parse and execute the code.
> > >
> > > First, omitting the reasoning conversation before queries prevents the Retriever from hallucinating based on redundant information. For instance, we observed cases where the Retriever, influenced by the Reasoner’s initial task breakdown analysis, attempted to solve the entire problem rather than focusing on the current query. Since initial analyses are incomplete without further information, such behavior often leads to failure. Second, the Reasoner is less prone to hallucinations without processing the retrieval process. Without isolation, the Reasoner may capture the Retriever’s graph-loading code and attempt to process the graph itself. This might lead to multiple reasoning steps in one code-writing attempt, effectively collapsing it into the SingleCoder variant tested in our ablation study. Additionally, we observed instances where the Reasoner relied on unreliable debugging outputs from the Retriever. For example, if the Retriever failed to fetch a room-door edge due to a formatting error, the Reasoner might incorrectly infer that the room has no neighbors.
> > >
> > > To mitigate these issues, our two-agent framework isolates retrieval and reasoning by manipulating the conversation history before invoking GPT, ensuring the intended behavior of LLMs. Unlike ReAct’s re-prompting approach, this framework effectively separates the roles for each agent, which we consider a non-trivial contribution.
> > >
> > > We also respectfully disagree with the assertion that the Python implementation lacks significant novelty, as it provides valuable insights for the LLMs + Scene Graph community. Traditionally, LLMs interpret graph structures directly in textual format, as reviewed in the manuscript's introduction and literature sections. In contrast, our work demonstrates that graph structures can be more accurately interpreted by processing them through code. We also show the compatibility of this insight with the LLMs with their graph schema understanding and code writing capacities, offering a novel formulation: treating graphs as databases rather than text corpora. This approach forms the foundation of our method and highlights its potential. Notably, even ReAct is applicable to this problem only under this new formulation, which further underscores the contributions of our work.
> > >
> > > We hope the above explanation clarifies your concerns, and we look forward to hearing your further thoughts.

---

> > > > ### Comment · Reviewer_Tptn · 2024-11-27
> > > >
> > > > Thank you for the detailed responses. However, I am still unclear about the key difference between ReAct and RwR.
> > > >
> > > > In RwR, the Reasoner is also reprompted with the entire conversation history, after which it generates the next  $a_t$ and $q_t$. The Retrieval agent (another agent) then processes $q_t$ and returns the subgraph to the Reasoner.
> > > >
> > > > Similarly, in ReAct, the entire conversation history is consumed to generate the next thought and action. The same agent then executes the action to retrieve information.  The action in ReAct is an API invocation, which is not influenced by the history.
> > > >
> > > > Any misunderstanding?

---

> > > > > ### Author Response · Authors · 2024-11-28
> > > > > **Further explanation on the two-agent v.s. one-agent framework (1)**
> > > > >
> > > > > Thanks for letting us know about your confusion. Here we explain the difference formally. We also prove the importance of this difference by conducting ablation study on the single-agent version of SG-RwR. At last, we show failure cases to qualitatively demonstrate the importance.
> > > > >
> > > > > Assume that a retrieval query $q_t$ demands the code writing action $c_t$ (let’s assume it also includes the code debugging attempts) to obtain the information $\mathcal{G}_t$, where $c_t$ and $\mathcal{G}_t$ is equivalent to the advanced version of the "action" and "observation" in ReAct. **The single-agent framework of the ReAct** utilizes iterative prompting, which means that each thought ($a_t$ and $q_t$) and action are derived from the past thoughts, actions, and observations:
> > > > >
> > > > > $a_t, q_t, c_t = f_{LLM}( \\{a_1, q_1, c_1, \mathcal{G}_1\\},  \\{a_2, q_2, c_2, \mathcal{G}_2\\}, ...)$
> > > > >
> > > > > In contrast, **in the two-agent framework of SG-RwR**, the Reasoner processes $\mathcal{G}_t$ directly without $c_t$, and the Retriever processes the query $q_t$ only without the analyses behind it:
> > > > >
> > > > > $a_t, q_t = f_{LLM}( \\{a_1, q_1, \mathcal{G}_1\\},  \\{a_2, q_2, \mathcal{G}_2\\}, ...)$
> > > > >
> > > > > $c_t = f_{LLM}( q_t)$
> > > > >
> > > > > The two-agent design is critical for the success of our method, due to the fact that the action space in our approach is free-form code-writing as opposed to the API selection as in ReAct. Without isolation of the retrieve and reason processes, both of them might be misguided by the redundant information. To confirm this, we conduct an additional **ablation study of the single-agent variant of SG-RwR**, where the analysis, query, and retrieval code is generated by processing the entire history as in ReAct ($1^{st}$ formula above), on the BabyAI tasks under the few-shot setting. We observe that, other than the obvious disadvantage of the increased token processing, **SG-RwR(single agent)** is also outperformed by the original SG-RwR by a significant margin. The results are shown in the table below:
> > > > >
> > > > > *(All evaluated under the few-shot setting)*
> > > > > |        | ReAct_trv | SG-RwR(Single-Agent) | SG-RwR  |
> > > > > |--------|-----------|----------------------|---------|
> > > > > | NumQ&A | 24%       | 90%                  | **95%** |
> > > > > | Trv1   | 62%       | 46%                  | **67%** |
> > > > > | Trv2   | 56%       | 31%                  | **61%** |

---

> > > > > ### Author Response · Authors · 2024-11-28
> > > > > **Further explanation on the two-agent v.s. one-agent framework (2)**
> > > > >
> > > > > The main reason is that, as explained in the previous response, the redundant information in the conversation history might mislead LLMs' generation process. Here we show a few failure cases of SG-RwR(Single-Agent) to prove this claim:
> > > > >
> > > > > **Case 1: The Retriever hallucinates in $c_t$ by generating a non-existent scene graph based on previously retrieved information $\mathcal{G}_1, \mathcal{G}2, ..., \mathcal{G}{t-1}$, instead of loading the correct scene graph.**
> > > > >
> > > > > *[Snippet of the hallucination in the retrieval Code]*
> > > > > ```
> > > > > # **Hallucination! Retriever hallucinates a scene graph containing only the previously retrieved items, instead of loading the real graph**
> > > > > # expected: G = load_sg(SG_FILENAME)
> > > > > scene_graph = {
> > > > >     "nodes": [
> > > > >         {"type": "root", "id": 0},
> > > > >         {"type": "room", "coordinate": [0, 0], "size": [8, 8], "id": 37},
> > > > >         {"type": "room", "coordinate": [8, 0], "size": [8, 8], "id": 23},
> > > > >         {"type": "room", "coordinate": [0, 8], "size": [8, 8], "id": 45},
> > > > >         {"type": "door", "color": "red", "coordinate": [4, 0], "is_locked": False, "id": 18},
> > > > >         {"type": "door", "color": "blue", "coordinate": [0, 4], "is_locked": False, "id": 13},
> > > > >         {"type": "ball", "color": "green", "coordinate": [9, 1], "id": 50},
> > > > >         {"type": "ball", "color": "yellow", "coordinate": [1, 9], "id": 51}
> > > > >     ],
> > > > >     "links": [
> > > > >         [0, 37], [0, 23], [0, 45],
> > > > >         [37, 18], [37, 13],
> > > > >         [18, 23],
> > > > >         [13, 45],
> > > > >         [23, 50],
> > > > >         [45, 51]
> > > > >     ]*
> > > > > }
> > > > >
> > > > > # Convert the scene graph to a NetworkX graph
> > > > > G = ...
> > > > >
> > > > > # Find the balls in the rooms with IDs 23 and 45
> > > > > for room_id in room_ids: ...
> > > > > ```
> > > > >
> > > > > **Case 2: Retriever writes the code to implement the entire problem solving process, instead of the current query.** This removes the critical iterative solving process and reduces SG-RwR to the inferior SingleCoder variant, which has higher hallucination risk. Below example hallucinate in step 2.
> > > > >
> > > > > *[Snippet of the hallucination in the retrieval Code]*
> > > > > ```
> > > > > # [Task] find the color of the key in a room next to the room with 3 green ball
> > > > > # [Past queries and retrieved info] "List the rooms that contain 3 green balls": "52"
> > > > > # [Current analysis] I have identified the room with ID 52 that contains 3 green balls. Now, I need to find the rooms that are next to this room and check if they contain a key. If a key is found, I will retrieve its color.
> > > > > # [Current query] List the doors connected to room 52.
> > > > >
> > > > > # Step 1: Find the room that contains 3 green balls
> > > > > room_with_3_green_balls = None
> > > > > for node in G.nodes:
> > > > >      ......
> > > > >
> > > > >  # Step 2: Find the rooms connected to the room with 3 green balls
> > > > > # Hallucination! It hallucinates that the neighbor room nodes are directly connected, whereas they should just have a shared door node neighbor.
> > > > >  neighboring_rooms = []
> > > > >  for door in connected_doors:
> > > > >      if G.nodes[node]['type'] == 'room':
> > > > >          neighbors = list(G.neighbors(node))
> > > > >          if room_with_3_green_balls in neighbors:
> > > > >              neighboring_rooms.append(node)
> > > > >
> > > > >
> > > > >  # Step 3: Check if the neighboring rooms contain a key and retrieve the color of the key if found
> > > > >  key_colors = []
> > > > >  for room in neighboring_rooms:
> > > > >      .....
> > > > >
> > > > >  # Print the colors of the keys found in the neighboring rooms
> > > > >  print("Colors of the keys found in the neighboring rooms:", key_colors)
> > > > > ```
> > > > > **Case 3: The Reasoner "hallucinates and forgets," neglecting previously identified missing information or reasoning steps in the subsequent solving process.** This is potentially because of the increased gap between the queried information in the conversation due to the addition of the retrieval code, or that the retrieval code biases the sampling process of LLMs.
> > > > >
> > > > > *[Snippet of the hallucination in the simplified reasoning trace]*
> > > > > > **[Explanation (analysis)]** Next, we need to determine if there are any **obstacles** or **locked doors** between the agent and the yellow box.First, we need to know the rooms in which the agent and the yellow box are located.
> > > > > >
> > > > > > **[Query]** Which room contains the agent and the yellow box?
> > > > > >
> > > > > > **[Code]** ```for ...```
> > > > > >
> > > > > > **[Query]** Are there any **doors** between room 5 and room 12? If so, provide their details including their node IDs, colors, coordinates, and lock status.
> > > > > >
> > > > > > **[Code]** ```for ...```
> > > > > >
> > > > > > **[Query]** What is the coordinate and node ID of the red key?
> > > > > >
> > > > > > **[Code]** ```for ...```
> > > > > >
> > > > > > **[Summary]**
> > > > > >
> > > > > >1. Pick up the red key.
> > > > > >
> > > > > >2. Open the red door. (**Hallucination! Does not query for the obstacles to determine the traversability, even though this step was established in earlier analysis.**)
> > > > > >
> > > > > >3. ...

---

> > > > > ### Author Response · Authors · 2024-11-28
> > > > > **Further explanation on the two-agent v.s. one-agent framework (3)**
> > > > >
> > > > > We hope that the formula above clearly explains the difference between the two-agent and one-agent framework. And that the additional experiment results and qualitative examples demonstrate the importance of the two-agent design in our method. Feel free to reach out for any additional questions. **We will include the explanation of the difference between our method and ReAct, as well as the additional experiment results we show in the rebuttal, into the final version of our paper.**

---

> > > > > > ### Author Response · Authors · 2024-12-02
> > > > > > **Looking forward to your reply!**
> > > > > >
> > > > > > Dear Reviewer Tptn,
> > > > > >
> > > > > > We would like to thank you again for the suggestions and insights you provided. To address your concerns, we have formulated the two-agent design and its difference compared to the one-agent framework (ReAct), and have added ablation experiments and qualitative examples to highlight the importance of this design. We also clarified the contribution of our method within the context of scene graph reasoning, where ReAct cannot be straightforwardly applied without our novel formulation of scene graphs as the database retrievable via code.
> > > > > >
> > > > > > As the discussion period approaches the end, we would appreciate if you could review our response. Please feel free to reach out if you have any additional questions.
> > > > > >
> > > > > > Thanks,
> > > > > >
> > > > > > Authors

---

> > > > > > > ### Comment · Reviewer_Tptn · 2024-12-03
> > > > > > >
> > > > > > > Thank you for the clarification.

---

### Official Review · Reviewer_ipZT · 2024-11-01

**Soundness:** 3
**Presentation:** 3
**Contribution:** 2
**Rating:** 6
**Confidence:** 4

**Summary:**

To address the issue of LLMs lacking the ability to handle large graph inputs and solve complex spatial tasks, this work proposes $\textit{RwR}$, a framework for reasoning on scene graphs with LLMs. $\textit{RwR}$ adopts a Reason-while-Retrieve style. The reasoner decomposes the task into small subtasks. Based on the output of the reasoner, the retriever extracts relevant graph information. Instead of feeding the entire graph into LLMs, this pipeline enables $\textit{RwR}$ to process task-related subgraph information, reducing the hallucinations and window size limitations of LLMs. Additionally, $\textit{RwR}$ has access to external tools, which further improve the reasoning ability of LLMs on scene graphs. Experiments in multiple simulation environments prove that $\textit{RwR}$ exhibits impressive performance.

**Strengths:**

1. This work represents the first effort in reasoning with scene graphs using LLMs, which facilitates the study of scene graphs.
2. Instead of simply using graphs-as-text as input to LLMs, this work proposes the Reason-while-Retrieve framework, which combines the reasoning and retrieval phases together, addressing the problem that LLMs cannot handle large input graphs.
3. $RwR$ utilizes schema-based grounding and code-writing for graph information, reducing hallucinations and improving reasoning ability, which can be applied in other graph-LLM-related domains.

**Weaknesses:**

1. $RwR$ retrieves and reasons multiple times, which increases the reasoning time.
2. The graph schemas and reasoning tools are also manually designed for specific tasks, causing issues when transferring to new tasks.
3. You conducted a few-shot experiment; however, you should conduct various few-shot experiments.

**Questions:**

1. I would like to see analysis about the increased time caused by retrieving more than once, compared to Retrieve-then-Reason method.
2. I was also wondering if $RwR$ can be used with other commonly used GraphRAG datasets, such as ExplaGraphs, SceneGraphs, and WebQSP. If so, please conduct experiments to prove its applicability or provide analysis. If not, please state the reasons why.
3. Why you do not contain retrieve-then-reason method in your baseline?

---

> ### Author Response · Authors · 2024-11-20
>
> We kindly refer the reviewer to the general response for the following issues:
> 1. The analysis on the increased inference time due to the iterative conversation process.
> 2. The assumption on the availability of the scene graph schemas and reasoning tools.
> 3. Complete results on the few-shot settings.
> Below we clarify the other concerns.
>
> ## Q1) Can RwR be used with other commonly used GraphRAG datasets, such as ExplaGraphs, SceneGraphs, and WebQSP?
> Our method is capable of processing the 3D SceneGraphs dataset [1]. As a matter of fact, as explained in Section 3.1, we use the same graph structure in the BabyAI experiments.
>
> However, we believe that general RAG graphs, such as ExplaGraphs [2] and WebQSP [3] datasets, fall out of the scope of the paper. As explained in the comparison between our method and ReAct in the General Response, our approach is designed for the more structured scene graphs data, with well-organized semantic information types such as json format. By understanding the structure described by the schema, our method can flexibly and accurately aggregate necessary information across the graph via code-writing. For general knowledge graphs without such structure or format, adoption of our method requires reducing the retrieval code into local context aggregation. In this case, it falls back to the generic iterative baseline such as ReAct. Hence, our manuscript considers spatial reasoning and planning tasks on environmental layout scene graphs, rather than general Q&A task on knowledge or explanation graphs.
>
> [1] Armeni, Iro, et al. "3d scene graph: A structure for unified semantics, 3d space, and camera." Proceedings of the IEEE/CVF international conference on computer vision. 2019.
> [2] Saha, Swarnadeep, et al. "ExplaGraphs: An explanation graph generation task for structured commonsense reasoning." arXiv preprint arXiv:2104.07644 (2021).
> [3] Yih, Wen-tau, et al. "The value of semantic parse labeling for knowledge base question answering." Proceedings of the 54th Annual Meeting of the Association for Computational Linguistics (Volume 2: Short Papers). 2016.
>
> ## Q2) Why not contain any retrieve-then-reason method in the baselines?
> We apologize for the unclarity in the original submission. There is a retrieve-then-reason method baseline we tested, which is SayPlan [1] in Table 1. We adopt this baseline because it is specifically designed for the hierarchical scene graph structure [2] for the spatial layout representation, which is what we used in the BabyAI experiments. Other retrieve-then-reason methods cited in the paper, including ThinkOnGraph (ToG) and ReasonOnGraph (RoG), are devised for different types of graphs, such as knowledge graphs. Hence they are less straightforward to be adapted to our spatial reasoning tasks.
>
> Specifically, SayPlan follows a room-level retrieving strategy, where it examines the subgraphs rooted at each room until the task-relevant subgraph is identified. It then enters the reasoning phase. We follow the original paper to create 3 graph APIs for the retrieving process:
> - **collapse(G)**: Collapse a graph and only keep the top room-level nodes.
> - **expand(node_id)**: Expand a room node and reveal all nodes rooted from this room node.
> - **contract(node_id)**: Remove all nodes rooted from the specified room node.
>
> We omit the *verify\_plan(plan)* API from the original paper, as we don’t assume a graph simulator that is able to simulate and verify the generated plan.
>
> We show they are not as effective as our method in spatial reasoning problems as in Table 2. The main reason is that retrieving without non-trivial reasoning can lead to suboptimal task solving traces. For example, the NumQ&A task requires retrieving a target room next to a referent room, but it is hard to identify the target room without reasoning and finding the referent room first.
>
> In the updated manuscript, we add a clear statement that SayPlan is one of the retrieve-then-reason methods in Section 3, baseline paragraph, and add more details in the Appendix G.
>
> [1] Rana, Krishan, et al. "Sayplan: Grounding large language models using 3d scene graphs for scalable robot task planning." 7th Annual Conference on Robot Learning. 2023.
> [2] Armeni, Iro, et al. "3d scene graph: A structure for unified semantics, 3d space, and camera." Proceedings of the IEEE/CVF international conference on computer vision. 2019.

---

> ### Author Response · Authors · 2024-11-25
> **Looking Forward to Your Reply**
>
> Dear Reviewer ipZT
>
> Thank you for your time to read and review our paper. We have followed your suggestions to complete the experiment results and provide analysis on the inference computation. We also further discuss the value and the positioning of our work.
>
> As the discussion period approaches the end, we would appreciate if you could review our response. Please feel free to reach out if you have any additional questions.
>
> Thanks, Authors

---

> > ### Comment · Reviewer_ipZT · 2024-11-27
> > **Thanks for your response**
> >
> > Thanks for your response. I will keep my positive score.

---

> > > ### Author Response · Authors · 2024-12-02
> > >
> > > Thank you for your valuable feedbacks!

---

### Official Review · Reviewer_MeSM · 2024-11-04

**Soundness:** 3
**Presentation:** 3
**Contribution:** 1
**Rating:** 5
**Confidence:** 3

**Summary:**

This paper proposes the RwR (reason while retrieve) framework which equips LLM with scene graph schema and provides a code interface for the LLM to interact with the graph. This is in contrast with some prior works that input the whole graph into the LLM's context. The Reason part is an LLM capable of reasoning over the environment and asks the retriever queries to complete. The retriever executes queries on the underlying scene graph and responds to the reasoning system. The authors show that this strategy is effective by testing this on three tasks showing superior performance compared to methods such as chain-of-thought

**Strengths:**

* The paper is well-written and easy to follow.
* Some prompt design choices are interesting (such as preventing the model from assuming things by detecting the word "assumption" in the response).
* The method shows superior performance compared to CoT and other intrinsic reasoning baselines.

**Weaknesses:**

* The main weakness is the lack of novelty of this work: The introduced method seems to be a ReAct-style system + Tool-integrated reasoning (over graphs). As the authors mention in the related work section, incorporating knowledge retrieval from graphs has also appeared in prior works. The main novelty of this work seems to be the design of a Python interface for interacting with scene graphs + prompt designs.

* The method relies on the availability of a scene graph, which may not be available (e.g., a partially observable environment), or easily extractable.

[1] Yao et al., ReAct: Synergizing Reasoning and Acting in Language Models, ICLR 2023

**Questions:**

* Can the method be extended to partially observable environments? For instance by providing the retriever an API to also modify the scene graph?

---

> ### Author Response · Authors · 2024-11-20
>
> We kindly refer the reviewer to the general response for the following issues:
> 1. What is the novelty of the work compared to ReAct.
> 2. The overreliance on the availability of the scene graphs.
> Below we clarify the other concerns.
>
> ## Q1) Extension to partially observable environments.
> Our framework can naturally extend to partially observation environments. To show this, we create a partially observable BabyAI scene graph environment, where the visible scene graph only contains visited rooms in the past and the newly revealed room by specific actions (i.e. open a door). After each action is executed, the environment provides feedback in the form of the execution result (successful or failed) and the agent's updated location. It also updates the stored scene graph to reflect the revised visibility. Note that the scene graph is not directly sent to our method in the textual form, but rather is saved out for the retriever to access via code. This environment examines if the agent is able to handle dynamically changing scene graphs and process interaction feedback.
>
> We modify our method to generate BabyAI actions during the reasoning process, rather than after as in the manuscript. Within the iterative framework, we achieve this with the simplest approach, by only modifying the prompt of the Reasoner. In particular, we add an additional “Agent commanding action” message type, and constraint its format in the prompt below:
>
> > *Agent commanding action. Command the agent to execute an action. Format:*
> >
> > *[Explanation]*
> >
> > *Explain in one sentence why executing this action.*
> >
> > *[Action]*
> >
> > *Action for the agent to execute. Must be one of the Agent Actions. e.g. pickup(10)*
>
> We also change the system prompt to make the Reasoner realize that it is controlling an agent:
>
> >*You are a graph-based planning assistant that can reason on a scene graph representing an environment, and control an agent to solve the tasks in the environment.*
>
> We test the above variant of our method, named *SG-RwR\_dyn*, on the traversal tasks explained in the paper. Since collecting ground truth interaction and reasoning trace is non-trivial, we only consider the zero-shot settings. We collect the results in the table below, side-by-side with its performance on the static graph reported in the paper as reference. SG-RwR_dyn achieves superior performance compared to the static case, since it receives additional environmental feedback during the reasoning process. This suggests that our method is able to interact with the environment, process feedback, and handle dynamically changing scene graphs.
>
> |      | SG-RwR_dyn | SG-RwR |
> |------|---------|-----|
> | Trv1 | **76%** | 61% |
> | Trv2 | **58%** | 56% |
>
> Note that this is by no means the optimal way to adopt our method for the partially observable environment. But it shows the potential of our method in these settings. Future effort can explore advanced prompting structure and techniques, or designing the graph annotation approach, as suggested by Reviewer MeSM, to further enhance the task solving performance in the partially observable environments.

---

> > ### Comment · Reviewer_MeSM · 2024-11-22
> >
> > Thank you for your response and the added ablations and comparison with ReAct. So, according to your response, the difference between your method and react is (1) A more sophisticated API call (via a retriever with python code execution). (2) Disentangling the reasoner with retriever. Regarding the second contribution (which is a significant part of the method), the reasoning in your response regarding having both reasoner and retriever (as opposed to a single agent) is
> > > We observe that this design might mislead the LLM when sampling for the next retrieval action, as it might hallucinate that the necessary information is already retrieved.
> >
> > Can you point out to the ablation study you did on this claim? A common practice for interleaving code with natural language reasoning is to interleave them as code comments (e.g., see [1]), from which the authors could take inspirations in their future versions.
> >
> > [1] Lyu et al., Faithful Chain-of-Thought Reasoning, 2023
> >
> >
> > Overall, I find the added comparison with ReAct and method name changes to have improved the paper's positioning (for instance replacing RwR with SG-RwR is a positive step), however, the paper still requires significant repositioning in terms of its contributions, for instance, the figure 1 should compare SG-RwR with methods such as ReAct to highlight its contributions (particularly (1) and (2) above), rather than other types of prompting strategies. As such, the paper requires a new round of submission and peer-review, and I maintain my current score.

---

> > > ### Author Response · Authors · 2024-11-24
> > >
> > > We appreciate your valuable feedback on our work. We would like to provide more details and explanations below, and update the manuscript to further clarify the positioning of our work.

---

> > > ### Author Response · Authors · 2024-11-24
> > > **Details on the observation that ReAct might hallucinate with previously retrieved information**
> > >
> > > Here we provide a detailed qualitative example. To shorten the response, some "thought" messages from the LLMs have been omitted and replaced with summaries of the intent behind subsequent actions, indicated by comments prefixed with '#':
> > > > …
> > > >
> > > > \# checking room 54 for 4 red balls
> > > >
> > > > **action**: get_attrs(54)
> > > >
> > > > **action**: get_neighbors(54)
> > > >
> > > > **action**: get_attrs(...)
> > > >
> > > > \#  checking room 34 for 4 red balls.
> > > >
> > > > **action**: get_attrs(34)
> > > >
> > > > …
> > > >
> > > > \#  checking room 27 for 4 red balls.
> > > >
> > > > **action**: get_attrs(34)
> > > >
> > > > …
> > > >
> > > > \#  **Hallucinate the neighbor relationship from the information retrieved for identifying 4 red balls**
> > > >
> > > > **thought**: I have found three red balls in room 27.  *Room 27 is adjacent to room 54 and 34* (**Wrong!**). Now I need to check for boxes in these adjacent rooms...
> > > >
> > > > **action**: *get_neighbors(54)*
> > >
> > > Since ReAct reprompts the entire past conversation, including intermediate retrieval steps and outcomes, back to the single agent, it risks hallucinating when generating the next step. In this example, it incorrectly assumes that the target adjacent rooms are a subset of previously retrieved rooms without performing additional retrieval to detect unchecked rooms or validate the connection by doors. In contrast, our two-agent design ensures that the Reasoner receives only the specifically queried information (e.g., "room 27" for the query "Find the room that contains 4 gray keys") without unnecessary context, thereby reducing hallucinations in subsequent generation steps.

---

> > > ### Author Response · Authors · 2024-11-24
> > > **Related work that interleaves code with natural language reasoning is to interleave them as code comments**
> > >
> > > We thank the reviewer for highlighting this common practice and providing a cited example, which offers valuable inspiration for our design. Regarding methodology, the approach in [1] is comparable to our SingleCoder variant, ablated in Table 3, as both are non-iterative methods that generate a single piece of code to solve the entire problem. The difference is that [1] introduces an additional language-based task break-down step, which assists the code-writing process, with the reasoning steps documented as comments. Despite the difference, our experiments with SingleCoderdemonstrate that generating a single piece of code is insufficient for addressing complex tasks, such as the traversal task, which demands sequential reasoning. Without access to intermediate results, SingleCoder attempts to write extended code to provide a global solution that accounts for all possible conditions (e.g., whether a door is open or closed, corresponding keys are present or absent, etc.). This often results in errors or early termination in the code, with "No Solution" outputed for unsolved conditions. In contrast, iterative methods like ours solve problems auto-regressively, generating the next piece of code based on previous reasoning results and thus avoiding the need to account for all unpresented scenarios.
> > >
> > > Nevertheless, [1] provides additional evidence that code-writing and tool usage can enhance the faithfulness and accuracy of problem-solving, which aligns with and supports our design choices. **We have added a citation to this paper in the relevant part in the Introduction.**
> > >
> > > [1] Lyu et al., Faithful Chain-of-Thought Reasoning, 2023

---

> ### Author Response · Authors · 2024-11-24
> **Further clarification on the position of our work with respect to interleaved methods such as ReAct**
>
> Although the problem scope of these methods are beyond the spatial tasks on scene graphs, we agree that our paper should include a discussion of existing interleaved retrieve-and-reason literature to better highlight the distinctions. In response, **we add a new paragraph in the Introduction section to cite and summarize related research, including ReAct, from this category [1-5]**. These methods, designed for reasoning over text corpora, rely on lexical or embedding-based retrieval to obtain additional information semantically related to previously generated sentences. For instance, as shown in [4], when answering "In what country was Lost Gravity manufactured?", the system retrieves paragraphs about Lost Gravity, identifies the constructing company, and subsequently retrieves relevant details about the company. However, we hypothesize that a direct application of these methods to scene graph tasks is unsuitable, as queries often require not only semantic understanding but also spatial reasoning based on the graph structure. For example, “find the room containing 4 red balls” needs understanding not only the color and type node attributes, but also the containing relationship based on the graph edges. This hypothesis is confirmed by the superior performance of our method against ReAct shown in the general response.
>
> We respectfully disagree with the comment that Figure 1 should compare with ReAct-like methods rather than the whole graph prompting or Retrieve-and-Reason methods.  **Our work is positioned within the domain of spatial reasoning on scene graphs, rather than the general RAG reasoning.**. As explained in the Introduction, all methods depicted in Figure 1 and benchmarked in Table 1 have been experimented on graph-structured data [6, 7]. Both strategies in Figure 1 have been previously applied to scene graphs [8, 9]. While interleaved retrieve-and-reason methods like ReAct are effective in other domains, it is not reasonable to assume they can be trivially adapted for spatial reasoning on scene graphs. As explained in the previous paragraph, our work introduces the first iterative method specifically designed to enhance LLMs performance on scene graph tasks, and we show that it outperforms the naive adaption of ReAct. This highlights our contribution.
>
> We sincerely hope this response, with additional explanation on the broader interleaved retrieve and reason literature as well as the problem scope considered by the paper, addresses your concerns and clarifies the position of our work.
>
> [1] Yao, Shunyu, et al. "React: Synergizing reasoning and acting in language models." arXiv preprint arXiv:2210.03629 (2022).
>
> [2]Jiang, Zhengbao, et al. "Active retrieval augmented generation." arXiv preprint arXiv:2305.06983 (2023).
>
> [3] Shao, Zhihong, et al. "Enhancing retrieval-augmented large language models with iterative retrieval-generation synergy." arXiv preprint arXiv:2305.15294 (2023).
>
> [4] Trivedi, Harsh, et al. "Interleaving retrieval with chain-of-thought reasoning for knowledge-intensive multi-step questions." arXiv preprint arXiv:2212.10509 (2022).
>
> [5] Press, Ofir, et al. "Measuring and narrowing the compositionality gap in language models." arXiv preprint arXiv:2210.03350 (2022).
>
> [6] Wang, Heng, et al. "Can language models solve graph problems in natural language?." Advances in Neural Information Processing Systems 36 (2024).
>
> [7] Sun, Jiashuo, et al. "Think-on-graph: Deep and responsible reasoning of large language model with knowledge graph." arXiv preprint arXiv:2307.07697 (2023).
>
> [8] Rana, Krishan, et al. "Sayplan: Grounding large language models using 3d scene graphs for scalable task planning." CoRR (2023).
>
> [9] Gu, Qiao, et al. "Conceptgraphs: Open-vocabulary 3d scene graphs for perception and planning." 2024 IEEE International Conference on Robotics and Automation (ICRA). IEEE, 2024.

---

> ### Author Response · Authors · 2024-11-25
> **Looking forward to your reply!**
>
> Dear Reviewer MeSM,
>
> We would like to thank you again for the suggestions you provided. We have included additional example outputs and analyses for the concerns you raised. We also explained more on the details of the two-agent framework as well as the novelty and value of approach for the LLMs + Scene Graphs area in the discussion with the Reviewer Tptn, which is closely relevant to the questions you raised.
>
> As the discussion period approaches the end, we would appreciate if you could review our response. Please feel free to reach out if you have any additional questions.
>
> Thanks,
>
> Authors

---

> > ### Author Response · Authors · 2024-12-02
> > **Looking forward to your reply!**
> >
> > Dear Reviewer MeSM,
> >
> > We would like to thank you again for the suggestions and insights you provided. Following them, we have clarified more on the position of our work (the focus on the scene graph reasoning), explained the difference between our method and ReAct (two-agent v.s. one-agent, code-writing v.s. API invoking), formulated the two-agent design, and added ablation experiments and qualitative examples to justify the design.
> >
> > As the discussion period approaches the end, we would appreciate if you could review our response. Please feel free to reach out if you have any additional questions.
> >
> > Thanks,
> >
> > Authors

---

### Author Response · Authors · 2024-11-20
**General Response**

We gratefully thank all reviewers for their careful reading and thorough feedback. **Here we provide clarifications and additional experiment results on shared concerns**. What’s more, to address reviewer-specific questions, we add **additional discussions and experiments in the individual response**. The revised contents in the new manuscript submission are highlighted with red.

---

> ### Author Response · Authors · 2024-11-20
> **Q1) Clarify the novelty compared to the ReAct method. [MeSM, Tptn, chDs]**
>
> We apologize for the confusion on novelty, due to the missing discussion about ReAct in the original submission. The tasks considered in the paper center on spatial reasoning on scene graphs, a focus that differs from the primary application domains of ReAct that are explored in previous literature, leading to our oversight on the similarity in the iterative design. Nonetheless, we believe there are key differences between our method and ReAct.  Below, we explain the difference between ReAct and the proposed method, and experiment the ReAct on the spatial reasoning tasks in the paper.
>
> ReAct [1] is an iterative retrieval and reason approach for the generic data source, where the retrieval is done by letting the LLM choose the information access API from the data source. In this paper, we consider the more structured scene graph data, where the semantic information is organized into categories and can be accessed through the corresponding labels. For the BabyAI scene graph example in the paper, the information of an item is organized in json format, with keys such as _“color”_ and _“location”_.  Our method is tailored to the well-structured data as described above. Contrary to ReAct, our information retrieval is done via code-writing, which not only composes basic data operations, but also performs logical filtering (such as for loop or condition checking as shown in Figure 2.) on the obtained information. Consequently, we obtain information with a smaller semantic gap compared to the reasoning requirement, which better facilitates the reasoning process.
>
> Another difference is that ReAct is an one-agent framework, where the new thoughts and retrieved information is continuously re-prompted to the same LLM agent. We observe that this design might mislead the LLM when sampling for the next retrieval action, as it might hallucinate that the necessary information is already retrieved. On the other hand, our approach is a two-agent framework that disentangles the retriever from the reasoning. When triggered, the retriever only receives the one sentence query, as explained in the Section. Thus, it is not biased by the conversation history. We will show examples later in this response.
>
> To better illustrate our points, we test ReAct on NumQ&A and Traversal tasks in BabyAI, where the scene graph is stored as Networkx data. We wrap 4 basic Networkx graph operations into APIs to be called by ReAct, which are detailed in the Appenidx G in the revised manuscript. To show the importance of the external programmatic tools in spatial reasoning, we also test a variant of ReAct, named _ReAct\_Trv_, whose action space includes the _traversal\_room_ function in Section 3.2. We provide both variants with few-shot examples, following the original implementation.The results are collected in the table below:
>
> |        | ReAct (few-shot) | ReAct_trv (few-shot) | **RwR-FS** |
> |--------|-------|-----------|------------|
> | NumQ&A | 24%   | 24%       | **95%**    |
> | Trv-1  | 13%   | 62%       | **67%**    |
> | Trv-2  | 0%    | 56%       | **61%**    |
>
> As shown in the table, our method outperforms either ReAct variant in all three tasks.
> Notably, our method outperforms ReAct in NumQ&A task by over 70%. We observe that on this task with a bigger scene graph (due to the larger environment as shown Figure 3 in the manuscript), ReAct is ineffective in finding task-relevant information. To address a simple query such as “which room contains 4 yellow balls”, ReAct might need to call dozens of APIs to aggregate information from the graph. On the other hand, ours might only need one round of retrieval query, as it is able to directly filter on the spatial relationships conveyed through the graph structure, as shown in the examples in Appendix E. Moreover, when solving the reasoning step such as “find the neighbor rooms of the room XX”, ReAct might attempt to generate the solution from the previously retrieved rooms, even though there might be more neighbors that have not been retrieved yet, leading to the wrong solution in some cases. Our method is able to avoid this situation, as the Retriever processes each query independent from previous conversations.
>
> In the revised manuscript, we add the ReAct baseline in the experiments in Section 3 and Table 1, and implementation details in Appendix G. We will include more discussion from the rebuttal responses in the camera ready version, if the paper were to be accepted. What’s more, **we remove RwR (retrieve-while-reason) from the manuscript title, and change the method abbreviation from RwR to SG-RwR throughout the paper**. This helps clarify the contribution we make in the paper.
>
> [1] Yao, Shunyu, et al. "React: Synergizing reasoning and acting in language models." arXiv preprint arXiv:2210.03629 (2022).

---

> > ### Author Response · Authors · 2024-11-24
> > **Continue on Q1) Clarify the novelty compared to the broader interleaved reason and retrieve methods. [MeSM, Tptn, chDs]**
> >
> > In response to the comment by the Reviewer MeSM, we would like to further clarify the position of our work comparing to the broader interleaved reasoning and retrieving literature. **The discussion is added as a separate paragraph in the Introduction section.**
> >
> > Existing iterative methods [1-5], including ReAct, are designed for reasoning over text corpora. They rely on lexical or embedding-based retrieval to obtain additional information semantically related to previously generated sentences. For instance, as shown in [4], when answering "In what country was Lost Gravity manufactured?", the system retrieves paragraphs about Lost Gravity, identifies the constructing company, and subsequently retrieves relevant details about the company.
> >
> > Unlike other prompting techniques compared against in the paper [6, 7], these methods have never been applied to scene graph tasks. We hypothesize that a direct application of these methods to scene graph tasks is unsuitable, as *queries often require not only semantic understanding but also spatial understanding from the graph structure*. For example, “find the room containing 4 red balls” needs understanding not only the "color" and "type" from node attributes, but also the containing relationship based on the graph edges. We then design a schema-informed code-writing-based iterative method specifically for reasoning on scene graphs. The hypothesis and the design is confirmed by the superior performance of our method against ReAct shown in the general response.
> >
> > [1] Yao, Shunyu, et al. "React: Synergizing reasoning and acting in language models." arXiv preprint arXiv:2210.03629 (2022).
> >
> > [2]Jiang, Zhengbao, et al. "Active retrieval augmented generation." arXiv preprint arXiv:2305.06983 (2023).
> >
> > [3] Shao, Zhihong, et al. "Enhancing retrieval-augmented large language models with iterative retrieval-generation synergy." arXiv preprint arXiv:2305.15294 (2023).
> >
> > [4] Trivedi, Harsh, et al. "Interleaving retrieval with chain-of-thought reasoning for knowledge-intensive multi-step questions." arXiv preprint arXiv:2212.10509 (2022).
> >
> > [5] Press, Ofir, et al. "Measuring and narrowing the compositionality gap in language models." arXiv preprint arXiv:2210.03350 (2022).
> >
> > [6] Wang, Heng, et al. "Can language models solve graph problems in natural language?." Advances in Neural Information Processing Systems 36 (2024).
> >
> > [7] Rana, Krishan, et al. "Sayplan: Grounding large language models using 3d scene graphs for scalable task planning." CoRR (2023).

---

> > > ### Author Response · Authors · 2024-11-29
> > > **Two-agent framework (SG-RwR) v.s. One-agent framework (ReAct): Formulation and Ablation Study [MeSM, Tptn, chDs]**
> > >
> > > We would like to explain more on the shared questions about the difference between the two-agent framework in SG-RwR and the one-agent framework in ReAct. We also show the ablation study as well as the failure cases of the one-agent variant of SG-RwR to highlight the significance of the two-agent framework in ReAct.
> > >
> > > ReAct generates "thought" and "action", and then receive "observation" as the execution result of the "action. In our method, the analysis $a_t$ and retrieval query $q_t$ are equivalent to "thought", the retrieval code writing, denoted $c_t$ (let’s assume it also includes the code debugging attempts), is equivalent to the advanced "action", and the retrieved graph information $\mathcal{G}_t$ is equivalent to the "observation". **The single-agent framework of the ReAct** utilizes iterative prompting, which means that each thought ($a_t$ and $q_t$) and action are derived from the past thoughts, actions, and observations:
> > >
> > > $a_t, q_t, c_t = f_{LLM}( \\{a_1, q_1, c_1, \mathcal{G}_1\\},  \\{a_2, q_2, c_2, \mathcal{G}_2\\}, ...)$
> > >
> > > In contrast, **in the two-agent framework of SG-RwR**, the Reasoner processes $\mathcal{G}_t$ directly without $c_t$, and the Retriever processes the query $q_t$ only without the analyses behind it:
> > >
> > > $a_t, q_t = f_{LLM}( \\{a_1, q_1, \mathcal{G}_1\\},  \\{a_2, q_2, \mathcal{G}_2\\}, ...)$
> > >
> > > $c_t = f_{LLM}( q_t)$
> > >
> > > The two-agent design is critical for the success of our method, due to the fact that the action space in our approach is free-form code-writing as opposed to the API selection as in ReAct. Without isolation of the retrieve and reason processes, both of them might be misguided by the redundant information. To confirm this, we conduct an additional **ablation study of the single-agent variant of SG-RwR**, where the analysis, query, and retrieval code is generated by processing the entire history as in ReAct ($1^{st}$ formula above), on the BabyAI tasks under the few-shot setting. We observe that, aside from the obvious drawback of the increased token processing, **SG-RwR(single agent)** is also outperformed by the original SG-RwR by a significant margin. The results are shown in the table below:
> > >
> > > *(All evaluated under the few-shot setting)*
> > > |        | ReAct_trv | SG-RwR(Single-Agent) | SG-RwR  |
> > > |--------|-----------|----------------------|---------|
> > > | NumQ&A | 24%       | 90%                  | **95%** |
> > > | Trv1   | 62%       | 46%                  | **67%** |
> > > | Trv2   | 56%       | 31%                  | **61%** |
> > >
> > > **For failure cases and more analysis, please refer to the discussion with Reviewer Tptn.** We hope the formulation and the ablation study clarify the difference and the importance of the two-agent design. **All explanation above on the difference between SG-RwR and ReAct will be added to the final version of our manuscript.**

---

> ### Author Response · Authors · 2024-11-20
> **Q2) Is the assumption of the availability of scene graphs, scene graph schema, and reasoning tools too strong? [ipZT, MeSM]**
>
> Although our work assumes the availability of the scene graph and schema, we argue that exploring LLMs with scene graphs remains highly valuable, as this approach has been utilized as an important submodule in embodied AI and robotics systems. As reviewed in the introduction and literature review, extensive research has focused on constructing scene graphs from perceptual inputs, from which both the scene graph and schema will be available. As a result, LLMs have been used with the scene graphs in robotic planning for large indoor environments [1][2].  Additionally, scene graphs and schemas are readily accessible in simulators like Unity, demonstrating their applicability for embodied AI. For instance, Octopus [3] leverages LLMs and scene graphs to label ground-truth actions automatically for training policy models. Similarly, gaming agents and other domains [4] can benefit from scene graph reasoning in simulated environments. These examples underscore the potential of scene graphs as a powerful environmental representation in the era of LLMs, which is the key motivation for our research.
>
> We also respectfully disagree with the reviewer [ipTZ] that those assumptions limit the generalization ability of our approach. For both the NumQ&A and traversal tasks, we use the same BabyAI scene graph schema and achieve competitive performance. Furthermore, we test our method in two different environments, BabyAI and VirutalHome, with distinct scene graph structures. While we manually curate the scene graph schema, the purpose here is to isolate and focus the research on the LLM-based scene graph processing module. As mentioned above, the design of the schema and the generation of the scene graph belong to a separate line of research.
>
> For the reasoning tools, we note that they are not an essential component of our system. This is evidenced by our method's superior performance on the NumQ&A and VirtualHome tasks without using any tools. Instead, we incorporate them to demonstrate our system’s ability to effectively leverage external tools for tasks beyond the inherent capacity of LLMs, and the importance of using tools for complex tasks. For example, in the traversal task shown in the manuscript, the reasoning tools significantly enhance performance. This benefit is universal for all LLM-based approaches with the tool-using design, such as the ReAct_trv shown in the last response question. This capacity highlights the potential of our system for more complicated applications leveraging advanced tools, such as policy models for embodied interaction or point cloud models for processing multimodality attributes from the graph.
>
> [1] Gu, Qiao, et al. "Conceptgraphs: Open-vocabulary 3d scene graphs for perception and planning." 2024 IEEE International Conference on Robotics and Automation (ICRA). IEEE, 2024.
> [2] Rana, Krishan, et al. "Sayplan: Grounding large language models using 3d scene graphs for scalable task planning." CoRR (2023).
> [3] Yang, Jingkang, et al. "Octopus: Embodied vision-language programmer from environmental feedback." European Conference on Computer Vision. Springer, Cham, 2025.
> [4] Hu, Sihao, et al. "A survey on large language model-based game agents." arXiv preprint arXiv:2404.02039 (2024).

---

> ### Author Response · Authors · 2024-11-20
> **Additional Analysis and Experiment Results**
>
> ## Q3) Analysis about the increased inference compute due to the iterative reasoning and retrieving process. [ipZT, chDs]
> We provide an analysis on the number of conversation rounds and processed tokens in **Appendix E and Figure 7**, while including the token counts for the CoT baseline and the original textualized graph for reference.
>
>  **Importantly, the computational cost of our method depends primarily on task complexity rather than graph size**. For the simpler NumQ&A task, which requires fewer reasoning steps, the average conversation iterations is 3. This is significantly fewer than the 8.15 needed for the more complicated traversal task. Additionally, the average processed tokens up to the average conversation iteration count are less than that of the graph size or the CoT input, thanks to the code-writing-based retrieval strategy that filters task-relevant information and avoids processing redundant information from the graph.
>
> Conversely, the complicated traversal task requires more iterations on average. Since the Reasoner processes cumulative conversation history, the processed token count grows beyond the graph size. However, this trade-off in compute cost yields superior performance, as demonstrated in Table 1. Conceptually, our method balances computational cost with performance for complex tasks, which is the primary focus of this work.
>
> While reducing computational cost is not our current focus, there is potential for further optimization. We observe occasional suboptimal reasoning traces caused by the stochasticity of LLM generation, leading to redundant information retrieval. For example, instead of querying directly for "the room with 3 yellow boxes" or "counts of yellow boxes in all rooms," the Reasoner might query for "all items in all rooms," resulting in a larger portion of the graph being processed. Future work could address this issue by labeling optimal reasoning traces as few-shot examples to guide and constrain LLM behavior, reducing unnecessary computation.
>
>
> ## Q4) More results on the few-shot settings. [ipZT,  chDs]
> We further evaluate the few-shot variants of our method, including both _SG-RwR-FS_ and _SG-RwR-A_, on the NumQ&A task. Similar to that on the 2D traversal task, we provide the Reasoner with ground truth task and solution examples _without_ detailed dialogic solving process.
>
> _SG-RwR-FS_ and _SG-RwR-A_ achieve 95% and 97%, respectively. This is equal to or slightly higher than the already near-perfect 95% performance of the zero-shot _SG-RwR_, surpassing other few-shot methods by more than 30%. This further validates our claim that our method is compatible with the task-level in-context learning, requiring no agent-level annotations.
>
> ## Correct CoT baseline result on NumQ&A
> We realize that, in the original submission, we mistakenly provide the CoT method with the 2D traversal demonstrations in the NumQ&A task. We sincerely apologize for this oversight and have rerun the experiment with the correct configuration. As a result, the CoT performance is updated from 45% to 53% in Table 1. Importantly, **the corrected result does not alter any existing conclusions**: our method remains significantly more effective for the NumQ&A task, outperforming the corrected CoT baseline by more than 40%, both with and without few-shot examples, as shown in the updated Table 1. Nonetheless, we would like to bring this correction to the reviewers’ attention for transparency

---

### Meta-Review · Area_Chair_L2w9 · 2024-12-16

**Metareview:**

The paper proposes a reason-while-retrieve framework for reasoning on scene graphs with LLMs. The main critique from the reviewers (MeSM, Tptn, and chDs) is the limited technical contribution compared to ReAct. Reviewer chDs is also concerned about the compute analysis the authors provided. The rebuttal did not directly or clearly address all these concerns. As a result, three out of four reviewers recommended rejection (6555). The AC agrees and recommends rejecting the paper.

**Additional Comments On Reviewer Discussion:**

The main critique from the reviewers (MeSM, Tptn, and chDs) is the limited technical contribution compared to ReAct. Reviewer chDs is also concerned about the compute analysis the authors provided. The rebuttal did not directly or clearly address all these concerns.

---

### Decision · Program_Chairs · 2025-01-22

Reject